# Structural basis of mammalian mucin processing by the human gut *O*-glycopeptidase OgpA from *Akkermansia muciniphila*

Beatriz Trastoy [1,4], Andreas Naegeli[2,4], Itxaso Anso [1,4], Jonathan Sjögren [2✉] & Marcelo E. Guerin [1,3✉]

*Akkermansia muciniphila* is a mucin-degrading bacterium commonly found in the human gut that promotes a beneficial effect on health, likely based on the regulation of mucus thickness and gut barrier integrity, but also on the modulation of the immune system. In this work, we focus in OgpA from *A. muciniphila*, an *O*-glycopeptidase that exclusively hydrolyzes the peptide bond *N*-terminal to serine or threonine residues substituted with an *O*-glycan. We determine the high-resolution X-ray crystal structures of the unliganded form of OgpA, the complex with the glycodrosocin *O*-glycopeptide substrate and its product, providing a comprehensive set of snapshots of the enzyme along the catalytic cycle. In combination with *O*-glycopeptide chemistry, enzyme kinetics, and computational methods we unveil the molecular mechanism of *O*-glycan recognition and specificity for OgpA. The data also contribute to understanding how *A. muciniphila* processes mucins in the gut, as well as analysis of post-translational *O*-glycosylation events in proteins.

[1] Structural Biology Unit, Center for Cooperative Research in Biosciences (CIC bioGUNE), Basque Research and Technology Alliance (BRTA), Bizkaia Technology Park, Building 801A, 48160 Derio, Spain. [2] Genovis AB, Box 790, 22007 Lund, Sweden. [3] IKERBASQUE, Basque Foundation for Science, 48013 Bilbao, Spain. [4] These authors contributed equally: Beatriz Trastoy, Andreas Naegeli, Itxaso Anso. ✉email: jonathan.sjogren@genovis.com; mrcguerin@gmail.com

The appropriate balance of the gut microbiota composition has a crucial impact on human health[1,2]. Perturbations of such equilibrium have been associated with the occurrence of several pathologies, including metabolic disease, cardiovascular disease, type-2 diabetes, cancer and inflammatory bowel disease[3–7]. Metagenomics sequencing of fecal samples revealed more than 1000 different bacterial species in the gut microbiota[8–10]. Each individual human is estimated to host at least 160 different species[11]. The gut microbiome is dominated by members of only two phyla, Firmicutes and Bacteroidetes[12,13]. Members of the phylum Proteobacteria are also frequently represented but much less prominently. The composition of the intestinal microbiome depends on the composition of the consumed diet[2,14–16]. Thus, the gut microbiota maintains a mutualistic relationship with the host: it provides bacteria with access to food, while bacteria contribute to the digestion of complex foods and the synthesis of essential metabolites to the host. Although the exceptional diversity of the host and dietary glycans, the human genome contributes with a small set of degradative enzymes capable of fully processing a small set of glycans containing only one or two different sugar linkages[17]. Strikingly, the gut symbiotic microbiota provides the complementary enzymatic machinery necessary to orchestrate the depolymerization of glycan structures into their sugar components that otherwise could not be processed by the host[18,19]. *Bacteroides thetaiotaomicron* represents a paradigm on how a prominent gut Bacteroidetes is able to catabolize dietary glycans[20,21]. The *B. thetaiotaomicron* genome contains polysaccharide utilization loci (PULs) that encode highly specific carbohydrate-active enzymes, surface glycan-binding proteins and transporters, with each PUL coordinating the degradation of a given glycan[22–24]. The gene products of PULs have been termed Sus-like systems because they are reminiscent of the starch utilization system (Sus), but harbor enzymes that are predicted to target non-starch glycans[25]. Sus-like systems represent up to one-fifth of the *B. thetaiotaomicron* genome, and are widespread among Bacteroidetes members[26–28]. They participate in the recognition and specificity for a broad range of plant and animal tissue glycans that are expected to enter the human gut, as well as the breakdown of *N*- and *O*-linked glycans[21,23,29–36].

*Akkermansia muciniphila* is a Gram-negative bacteria from the phylum Verrucomicrobia that accounts for 1–5% of large intestinal bacteria in healthy adults[37–39]. Several studies support the notion that *A. muciniphila* plays a crucial role in the mutualism relationship between the gut microbiota and the host, modulating the gut barrier function and other physiological and homeostatic roles during obesity and type 2 diabetes[2,40–42]. *A. muciniphila* is a mucin-degrading bacterium that promotes a beneficial impact on human health. This positive effect has been correlated with the regulation of mucus thickness and gut barrier integrity, but also with the modulation of the immune system by interaction with Toll-like receptor 2[43,44]. Mucins are the major organic components of the mucus layers which protect the epithelium of the gastrointestinal, respiratory, and urinary tract against pathogens and mechanical damage[45–47]. Mucins are a family of secreted or membrane-associated proteins heavily *O*-glycosylated, which can also be *N*-glycosylated, but much more sparsely[48]. The central part of mucins is formed of tandem repeats sequences of 8 to 169 amino acid residues rich in serine, threonine, and proline[49,50]. The serine and threonine tandem repeats serve as a scaffold for *O*-glycosylation, where one single mucin molecule can present hundreds of glycan structures and thereby create locally high concentrations of certain glycans[51]. The high glycan content gives mucins a considerable water-holding capacity and resistance to proteolysis, which are important properties of the mucosal protecting barrier[52]. The first step in the *O*-glycosylation biosynthetic pathway of mucins is the addition of *N*-acetylgalactosamine (GalNAc) from UDP-GalNAc to the hydroxyl groups in serine and threonine residues, a reaction catalyzed by a large family of up to 20 different polypeptide GalNAc-transferases[53–55]. The GalNAc residue can be further extended with other sugars, including galactose, *N*-acetylglucosamine, fucose or sialic acid, but not glucose, mannose or xylose residues[48]. Therefore, mucin *O*-glycans can be very heterogeneous, with hundreds of different chains present in some mucins. The *A. muciniphila* genome is expected to encode the apparatus to hydrolyze the peptide and glycosidic linkages to process mucins in the large intestine[56]. However, such information, as well as the precise mechanisms that *A. muciniphila* uses to process mucins in the gut, generating a positive impact on health, still remains a challenge.

Here we focus on OgpA from *A. muciniphila*, a recently discovered *O*-glycopeptidase that exclusively hydrolyzes the peptide bond *N*-terminal to serine or threonine residues substituted with an *O*-glycan[57]. OgpA and the catalytically inactive mutant OgpA$_{H205A/E206A}$ are currently used to analyze the *O*-glycosylation present in proteins and peptides. It is worth noting that the analysis of post-translational *N*-glycosylation events in proteins is well established and facilitated by a wide range of available enzymes. In contrast, the study of protein *O*-glycosylation remains a major challenge due to the lack of equivalent enzymes and the inherent structural heterogeneity of *O*-glycans[58,59]. OgpA and OgpA$_{H205A/E206A}$ are commercialized as OpeRATOR® and GlycOCATCH®, respectively, for sample preparation of *O*-glycosylated biopharmaceuticals (Fig. 1a, b; Genovis AB., Sweden). The enzymatic activity of OgpA is used to generate *O*-glycan specific digestion patterns that can be used to localize mucin-type *O*-glycosylation sites and their glycoforms, whereas the catalytically inactive mutant OgpA$_{H205A/E206A}$ is used for enrichment of *O*-glycosylated proteins and peptides. The molecular mechanism by which OgpA specifically recognizes *O*-glycosylation sites remains to be defined. In this work, we provide the high-resolution structures of OgpA in (i) its unliganded form, (ii) in complex with the glycodrosocin GKPRPYSPRPT(Gal-GalNAc) SHPRPIRV peptide substrate (GD) and (iii) its product. In combination with site-directed mutagenesis, enzymatic activity, *O*-glycopeptide chemistry, and computational methods we unveil the molecular basis of substrate specificity and catalysis of OgpA, contributing to understand *O*-glycosylated mucin processing in *A. muciniphila*.

## Results

**The overall structure of full length OgpA.** OgpA from *A. muchiniphila* comprises 385 residues (OgpA$_{WT}$ hereafter; B2UR60, UniProt code; ACD04945.1, GenBank) with a predicted signal peptide (residues 1–24) that was removed from the construct for crystallization purposes. The crystal structure of full-length OgpA in its unliganded form was solved using zinc single-wavelength anomalous dispersion (Zn-SAD). A first crystal form of OgpA was obtained in the tetragonal *P* 4$_1$ 2$_1$ 2 space group at 1.9 Å resolution (OgpA$_{WT1}$; PDB code 6Z2D; Fig. 1c–e; Supplementary Figs. 1 and 2; Supplementary Table 1 and "Methods" section). A second crystal form of OgpA was obtained in the orthorhombic *P* 2$_1$ 2$_1$ 2$_1$ space group at 1.6 Å resolution (OgpA$_{WT2}$; PDB code 6Z2O; Supplementary Table 1). The high quality of the electron density maps allowed the trace of residues 27–380 (OgpA$_{WT1}$) and 25 to 382 (OgpA$_{WT2}$; Fig. 1c–e; Supplementary Fig. 2). The overall protein scaffold and the conformation of the OgpA$_{WT1}$ and OgpA$_{WT2}$ were essentially preserved (root mean square deviation, r.m.s.d., of 0.37 Å for 350 residues). However, the connecting loop α4–α5 (loop 9; residues 227–246) is unstructured in OgpA$_{WT2}$ whereas in OgpA$_{WT1}$

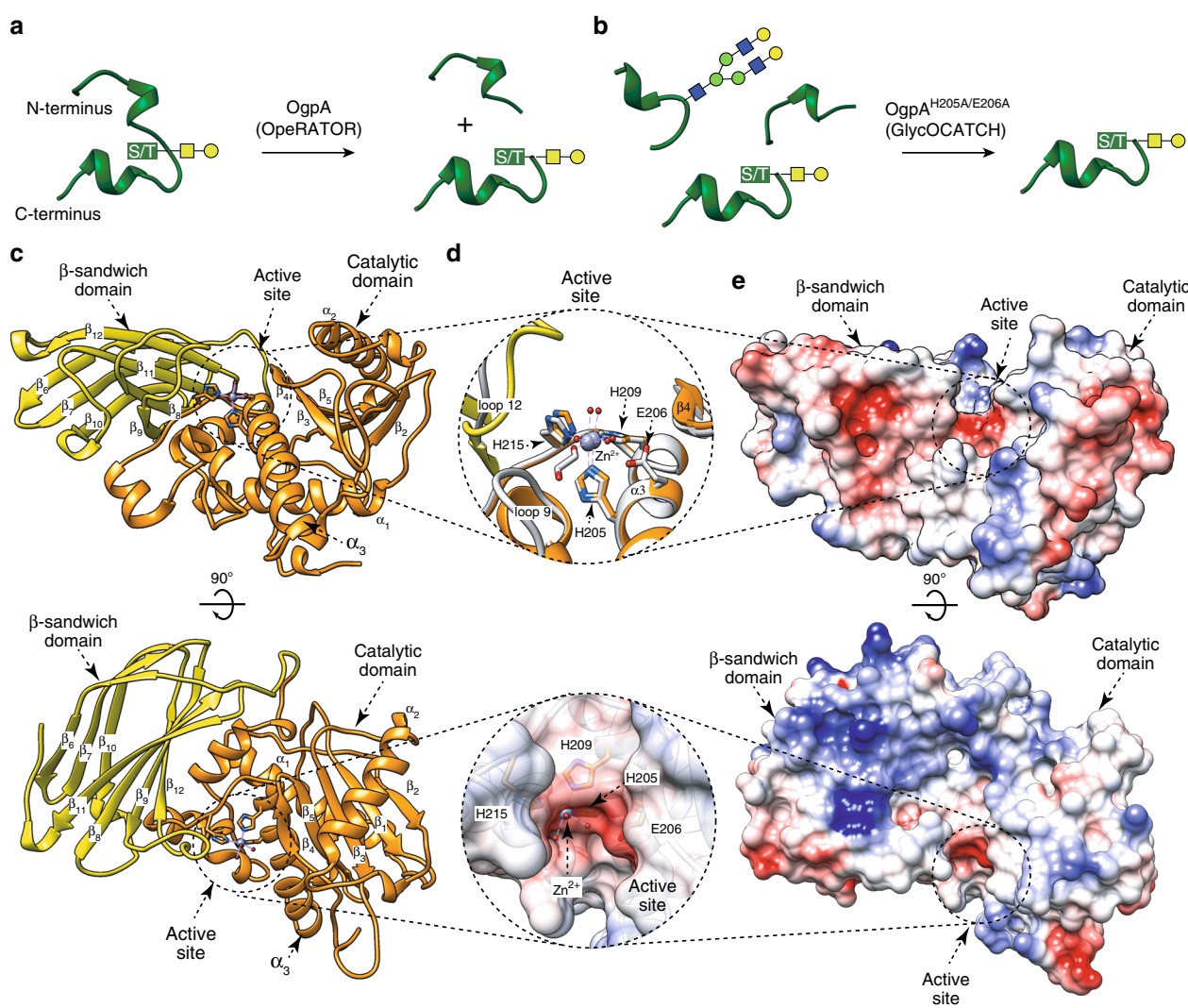

**Fig. 1 The overall structure of OgpA. a** Schematic representation of OpeRATOR (OgpA) and **b** GlycOCATCH activities on *O*-glycopeptides. **c** Two cartoon representations showing the general fold and secondary structure organization of OgpA$_{WT1}$, including the catalytic (orange) and β-sandwich (yellow) domains. We have decided to show the OgpA$_{WT1}$ structure because the density map explains the architecture of all loops that decorate the active site **d** Upper panel. Close up view of the active site of OgpA shown as cartoon/stick representation of the superimposed OgpA$_{WT1}$ (yellow and orange) and OgpA$_{WT2}$ (gray) structures. Lower panel. Close up view of the active site of OgpA shown as electrostatic surface representation. **e** Two electrostatic surface representations of OgpA$_{WT1}$ showing the location of the putative *O*-glycopeptide binding site and the catalytic site.

displays a short α-helix. Furthermore, we were unable to fully model the β7–β8 connecting loop (loop 12; residues 309–320) on OgpA$_{WT2}$ crystal structure because it is disordered. We have decided to use the OgpA$_{WT1}$ crystal form for our descriptions. A close inspection of the two crystal structures revealed that the protein crystallized as a monomer (Supplementary Fig. 1). OgpA$_{WT}$ comprises two domains from the N- to the C-terminus: (i) a catalytic domain (residues 25–262) followed by (ii) a β-sandwich domain (residues 263–385). The central core of the N-terminal domain consists of a five-stranded β-sheet with topology β2–β1–β3–β5–β4 (β4 is antiparallel), surrounded by three long α-helices, α1, α2, and α3, with an overall size of $43 \times 42 \times 42$ Å (Fig. 1c). The first β-sheet of the antiparallel β-sandwich domain consists of four β-strands with topology β6–β7–β8–β11, whereas the second β-sheet comprises four β-strands with topology β13–β12–β9–β10, with an overall size of $39 \times 28 \times 19$ Å (Fig. 1c–e; "Methods" section). Both domains are separated by a long and open groove that runs parallel to the protein surface where the active site is located (Fig. 1e). Interestingly, a Zn$^{2+}$ atom coordinates with the side chains of three histidine residues, H205,

H209, and H215, a highly conserved triad among metalloproteases (Fig. 1d). A glutamate residue, E206, is well-positioned to act as a general base/acid during catalysis. In addition, M231, lies within a Met-turn providing the hydrophobic base for the Zn$^{2+}$-binding site (Supplementary Fig. 3)[60–63]. The OgpA catalytic domain is annotated as an M11 peptidase, also called the gametolysin peptidase M11 family, according to the NCBI database. However, there are no *Akkermansia* sp. entries for the M11 family in MEROPS, a database that uses a hierarchical, structure-based classification of peptidases[64]. Interestingly, amino acid sequence homology analysis using the MEROPS scan dataset (MEROPS-MPRO) indicates that the OgpA metal-binding motif shares high sequence identity with peptidases classified in the M12 family. Both families, M11 and M12, share the common motif HEXXHXXXXH, in which the three His residues are zinc ligands and the Glu residue has a catalytic function. Structurally, the closest homologs also belong to the M12 family. However, it is worth noting that the M11 family is not structurally characterized. The M11 and M12 families belong to the MA(M) subclan, which comprises a metzincin fold, with a methionine C-terminal

to the $Zn^{2+}$ atom[63,65]. Members of M11 and M12 families are proenzymes that require activation by limited proteolysis[63]. There is no experimental evidence to support that OgpA is a proenzyme. Taken together, all the data suggests that OgpA could be considered as the founding member of a new family of peptidases.

A search for structural homologs using the DALI server (see "Methods") revealed Zn-dependent metalloproteases with significant low structural similarity to OgpA catalytic domain: (i) TNF-alpha converting enzyme (TACE; PDB code 3G42; Z-score of 11.3; r.m.s.d. value of 3.5 Å for 171 aligned residues; 12% identity)[66] (ii) atrolysin C from *Crotalus atrox* (PDB code 1ATL; Z-score of 10.9; r.m.s.d. value of 3.5 Å for 164 aligned residues, 12% identity)[67], and (iii) *Ba*P1 from *Bothrops asper* (PDB code 1ND1; Z-score of 10.9; r.m.s.d. value of 3.4 Å for 163 aligned residues, 14% identity)[68]. The three proteins are classified in the MEROPS M12 family (Supplementary Fig. 4a–c) and do not show specificity against *O*-glycopeptides. In addition, we found the following structural homologs for the β-sandwich domain using the same server: (i) the fibronectin type III (FnIII) domain of SleM from *Clostrium perfringens* (PDB code 5JIP; Z-score of 8.8; r.m.s.d. value of 2.8 Å for 90 aligned residues; 17% identity)[69], (ii) vacuolar protein sorting-associated protein 26 A (VPS26A) (PDB code 6H7W; Z-score of 9.1; r.m.s.d. value of 2.1 Å for 95 aligned residues; 14% identity), and (iii) β-sandwich domain of glycoside hydrolase XacMan2A from *Xanthomonas axonopodis pv. citri* (PDB code 6BYI; Z-score of 9.0; r.m.s.d. value of 2.8 Å for 99 aligned residues; 11% identity; Supplementary Fig. 4d–f)[70]. SleM is a peptidoglycan lysin, composed of an N-terminal catalytic domain similar to the GH25 family lysozymes and a C-terminal FnIII domain. The latter is involved in the formation of the SleM dimer[69].

**The structure of OgpA in complex with *O*-glycan substrate**. To obtain the crystal structure of the enzyme-substrate complex, we used a catalytically inactive version of OgpA, in which the residues H205, that coordinates the $Zn^{2+}$ atom, and E206, the base/acid residue of the reaction, are mutated to alanine ($OgpA_{H205A/E206A}$)[60,68]. For more information about the catalytic mechanism of OgpA, please see Supplementary Note 1 and Supplementary Fig. 3. The crystal structure of $OgpA_{H205A/E206A}$ in complex with the *O*-glycopeptide glycodrosocin ($_1$GKPRPYSPRPT (Gal-GalNAc)SHPRPIRV$_{19}$; GD hereafter) was solved by molecular replacement methods ($OgpA_{H205A/E206A}$-GD-SUB; PDB code 6Z2P; Fig. 2a, b; Supplementary Figs. 1 and 2; Supplementary Table 1 and "Methods" section). $OgpA_{H205A/E206A}$-GD-SUB crystallized in the tetragonal*I*4 space group and diffracted to a maximum resolution of 2.16 Å. The overall protein scaffold and the conformation of $OgpA_{H205A/E206A}$-GD-SUB were essentially preserved with respect to the $OgpA_{WT1}$ crystal structure (r.m.s.d. of 0.47 Å for 353 residues). The *O*-glycopeptide $_4$RPYSPRPT(Gal-GalNAc)SH$_{13}$ substrate was unambiguously identified in the electron density map (Fig. 2c–e). We assume that the full length GD is in the crystal, but no density was observed for the residues 1–3 and 14–19, probably due to conformational flexibility. Strikingly, the GD substrate is inserted as an extra parallel β-strand, relative to the other four strands of the β-sheet comprised in the catalytic domain (Fig. 2f, g and Supplementary Fig. 5), in a binding pocket located in the cleft between the N- and C-terminal domains of OgpA (Fig. 2c). The *O*-glycopeptide is flanked by α$_3$ and β$_4$, as well as the connecting loops β$_2$–α$_2$ (loop 4; residues 107–119), β$_3$–β$_4$ (loop 6; residues 150–166), α$_3$–α$_4$ (loop 8; residues 211–219), α$_4$–α$_5$ (loop 9; residues 227–246) and β$_8$–β$_9$ (loop 12; residues 309–320; Fig. 2f, g). Amino acid residues located at both ends of the *O*-glycopeptide, including $_4$RPYS$_7$ and H$_{13}$, are solvent-oriented and do not interact with OgpA (Fig. 2b, c).

Interestingly, we observe an extensive network of interactions, primarily mediated by hydrogen bonds, between GD's $_8$PRPTS$_{12}$ main chain residues and OgpA's N315, H215, Y318, and Y236 side chain residues (Supplementary Figs. 5 and 6). The GD backbone forms hydrogen bonds with the M169 and Y167 main chains (Fig. 2f, g). In addition, P$_6$ side chain interacts with the N315 side chain, while the P$_8$ side chain makes hydrophobic interactions with the M169 and Y168 side chains. Finally, the R$_9$ side chain makes electrostatic interactions with the H209 and L213 main chains, and the S$_{12}$ side chain with N235, and hydrophobic interactions with the Y236 and Y318 side chains. OgpA mainly interacts with neighboring residues of the glycosylated threonine, suggesting that the enzyme shows a broad range specificity for the peptide backbone. In contrast, the disaccharide Gal-GalNAc is placed in a pocket decorated by four aromatic residues, Y116, F166, W199, and Y236 (Fig. 2f, g). Specifically, the GalNAc moiety is surrounded by F166, W199, and Y236. The O6 of GalNAc of GD makes a hydrogen bond with the K198 main chain and also interacts with the N235 side chain. The Gal residue is surrounded by Y116 (CH-π interaction), F166, and W199. Furthermore, the O6 of Gal makes hydrogen bonds with the V164 main chain and also interacts with the G163 backbone. Finally, the carbonyl oxygen of the 2-acetamido group of GalNAc forms a hydrogen bond with the O2 of the Gal residue.

Amino-acid sequence alignment revealed that OgpA shows a high degree of sequence identity with other putative metalloproteases from two phyla: Verrucomicrobia and Bacteroidetes. In addition to the proteases of other species of the genus *Akkermansia* of the phylum Verrucomicrobia, we found other homologs belonging to three classes of the phylum Bacteroidetes: Bacteroidia (*Marinifilum* and *Odoribacter*), Chitinophagia (*Chitonophaga*) and Flavobacteriia (*Elizabethkingia, Chryseobacterium, Zhouia* and *Imtechella*; Supplementary Fig. 7). The catalytic residue E206 and the residues that coordinate the $Zn^{2+}$ atom, H205, H209, and H215, are conserved among these species. Moreover, residues that interact with the disaccharide Gal-GalNAc, Y116, F166, K198, W199, N235, and Y236 in OgpA, are also conserved among the species, supporting a similar mechanism of *O*-glycan recognition (Supplementary Fig. 7). In the case of BN783_01361 from *Odoribacter* sp. *CAG:788*, the α$_3$ helix shows 5 residues less than the other homologs, suggesting that this putative metalloprotease could accept different *O*-glycopeptides than OgpA.

**The structure of OgpA in complex with *O*-glycan product**. The strategy for capturing a native binary enzyme-product complex was to perform co-crystallization experiments with full-length wild-type OgpA in the presence of the GD *O*-glycopeptide substrate. It is worth noting that the enzyme was active against GD ("Methods" section). Thus, we obtained a snapshot of OgpA in complex with the proteolysis product of GD ($OgpA_{WT}$-GD-PRO; PDB code 6Z2Q; Fig. 3; Supplementary Figs. 1 and 2; Supplementary Table 1 and "Methods" section). $OgpA_{WT}$-GD-PRO crystallized in the tetragonal space group $P\,4_1\,2_1\,2$ and diffracted to a maximum resolution of 2.34 Å (Supplementary Table 1). The overall architecture and the conformation of $OgpA_{WT}$-GD-PRO were essentially preserved with respect to the crystal structure of $OgpA_{WT1}$ (r.m.s.d. of 0.37 Å for 344 residues) and that of the $OgpA_{H205A/E206A}$-glycodrosocin complex (r.m.s.d. of 0.42 Å for 353 residues; Fig. 3a, b). The $_{11}$T(Gal-GalNAc)S$_{12}$ *O*-glycopeptide product was unambiguously identified in the electron density map (Fig. 3c–e). In contrast, there is no electron density for the $_1$GKPRPYSPRP$_{10}$ peptide, supporting the notion that it leaves the active site immediately after the proteolytic reaction takes place. For more information about the catalytic cycle of OgpA, please see Supplementary Note 1 and Supplementary Fig. 3. As depicted

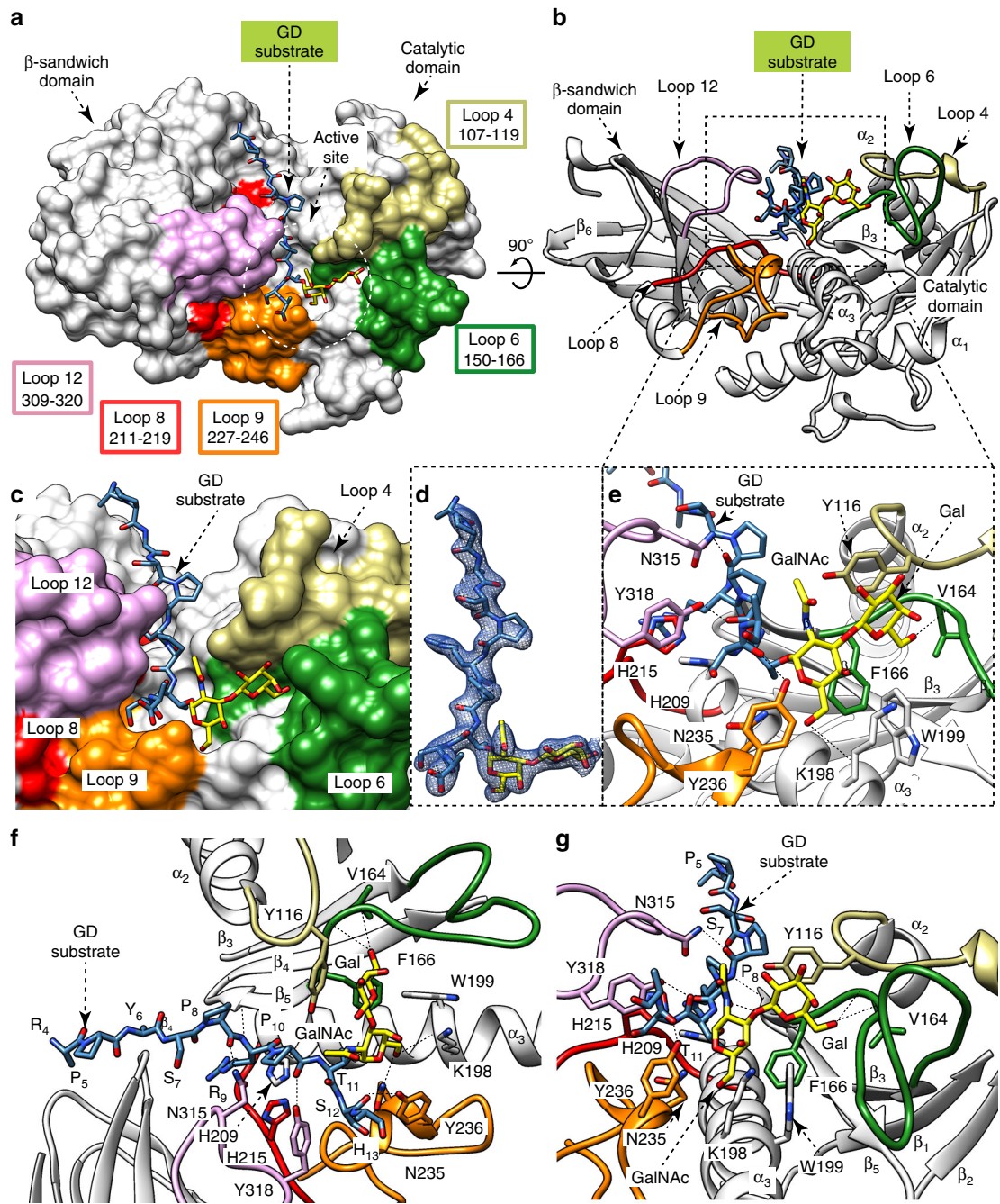

**Fig. 2 The *O*-glycopeptide substrate GD binding site. a** Surface representation of the OgpA$_{H205A/E206A}$-GD-SUB crystal structure, with annotated domains and loops. **b** Cartoon representation of the OgpA$_{H205A/E206A}$-GD-SUB crystal structure. **c** Surface representation of OgpA showing the location of the GD *O*-glycopeptide substrate into the active site. **d** Electron density map of the GD substrate shown at 1.0 σ r.m.s.d. **e–g** Three different cartoon representations of OgpA showing the location of the GD *O*-glycopeptide into the active site, the main residues and secondary structure elements. The key hydrogen bond interactions between OgpA and the GD-SUB are shown in dotted lines. The full list of interactions is reported in Supplementary Fig. 6.

in Fig. 3f, g, the conformation of the disaccharide Gal-GalNAc group observed in the $_4$RPYSPRPT(Gal-GalNAc)SH$_{13}$ substrate complex and in the $_{11}$T(Gal-GalNAc)S$_{12}$ product complex, superimpose very well, as well as all residues located at the substrate/product binding site.

**Structural basis of OgpA specificity for *O*-glycopeptides.** *O*-glycans are structurally diverse with up to 8 different core arrangements identified in mammals. We studied the substrate specificity of OgpA for the most common *O*-glycan core

structures in vitro using synthetic *O*-glycopeptides as substrates. To this end, we synthesized short *O*-glycopeptides based on the MUC1 sequence region, with a fluorescent label at the C-terminus. OgpA activity was measured using reverse-phase HPLC (Fig. 4a; Table 1; Supplementary Figs. 8 and 9). The peptide carrying a core 1 glycan (C1, Gal-β1,3-GalNAc) was by far the best substrate, while core 3 (C3, GlcNAc-β1,3-GalNAc) and α2,3-sialylated core 1 (3SC1, Neu5Ac-α2,3-Gal-β1,3-Gal-NAc), could be turned over although at very low rates. In contrast, there was no digestion of α2,6-sialylated core 1 (6SC1, Gal-β1,3-(Neu5Ac-α2,6)-GalNAc) or core 2 (C2, Gal-β1,3-(GlcNAc-

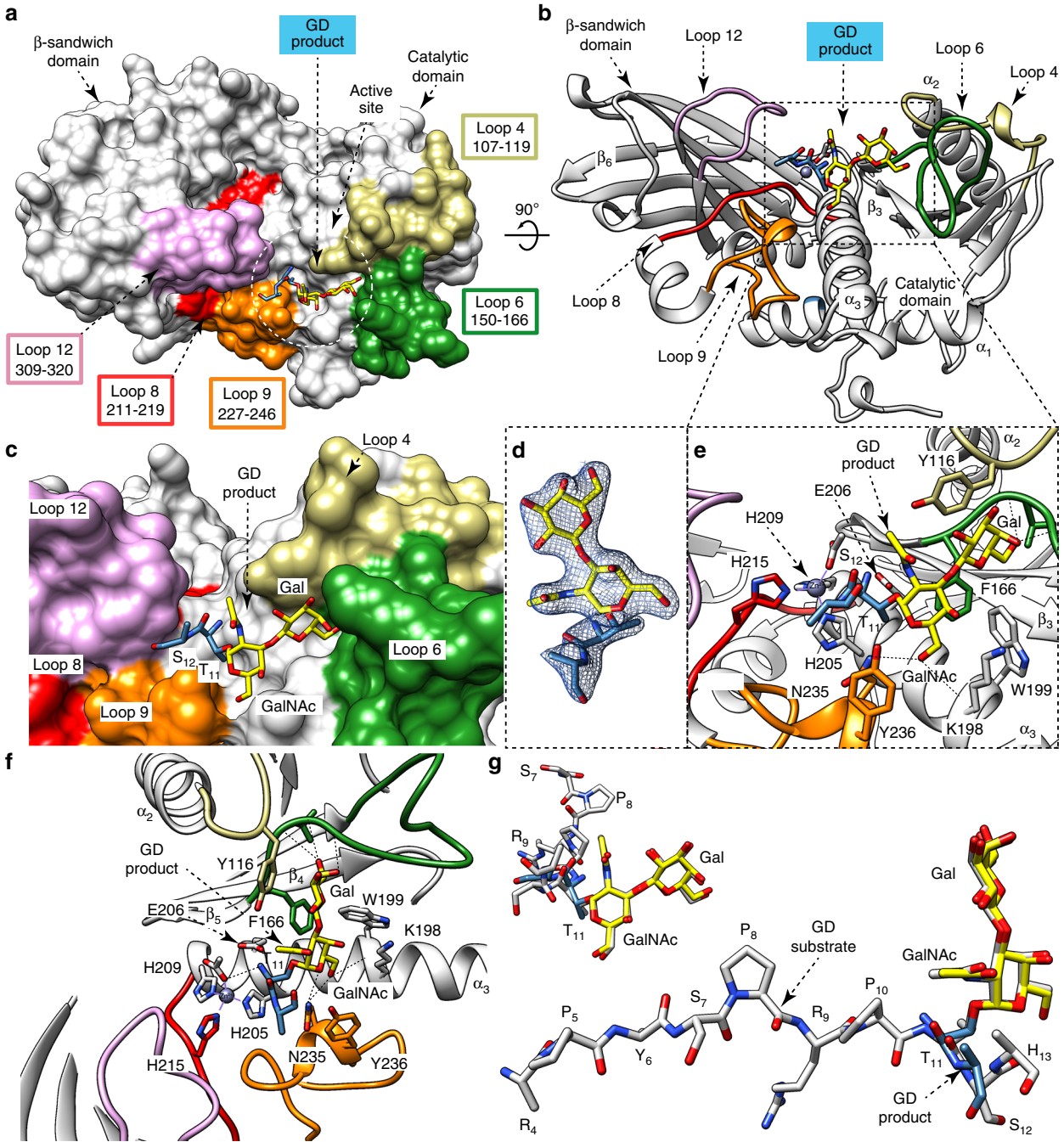

**Fig. 3 The *O*-glycopeptide GD product binding site. a** Surface representation of the OgpA$_{WT}$-GD-PRO crystal structure, with annotated domains and loops. **b** Cartoon representation of the OgpA$_{WT}$-GD-PRO crystal structure. **c** Surface representation of OgpA showing the location of the GD *O*-glycopeptide product into the active site. **d** Electron density map of the GD product shown at 1.0 σ r.m.s deviation. **e**, **f** Two different cartoon representations of OgpA showing the location of the GD *O*-glycopeptide product into the active site, the main residues and secondary structure elements. **g** Two views of the structural superposition of GD *O*-glycopeptide substrate (gray) and product (blue and yellow) in the OgpA$_{WT}$-GD-SUB and OgpA$_{WT}$-GD-PRO crystal structures, respectively. The key hydrogen bond interactions between OgpA and the GD-PRO are shown in dotted lines. The full list of interactions is reported in Supplementary Fig. 6.

β1,6)-GalNAc) *O*-glycopeptides. Peptides lacking the β1,3-linked galactose (Tn antigen, TnAg) and non-glycosylated peptides were not digested by OgpA.

To further understand the OgpA substrate specificity at the molecular level, we performed docking calculations with the natural C1 *O*-glycan substrate and the different *O*-glycan cores evaluated in the activity assays (Fig. 4, Table 1 and Supplementary Figs. 8 and 9). We first studied TnAg, which only displays the

GalNAc residue observed in C1. Although the TnAg can be placed in the binding pocket, it does not fulfil all of the requested interactions with the enzyme to be specifically recognized (Fig. 4c, Table 1). When 3SC1 is placed in the binding pocket of OgpA, the terminal Neu5Ac makes clashes with residues in the connecting loops between β2-α2 and β3- β4 (loops 4 and 6; Fig. 4d). Addition of a Neu5Ac residue to the GalNAc of C1, in the forms of Gal-β1,3-(Neu5Ac-α2,6)-GalNAc (6SC1) and

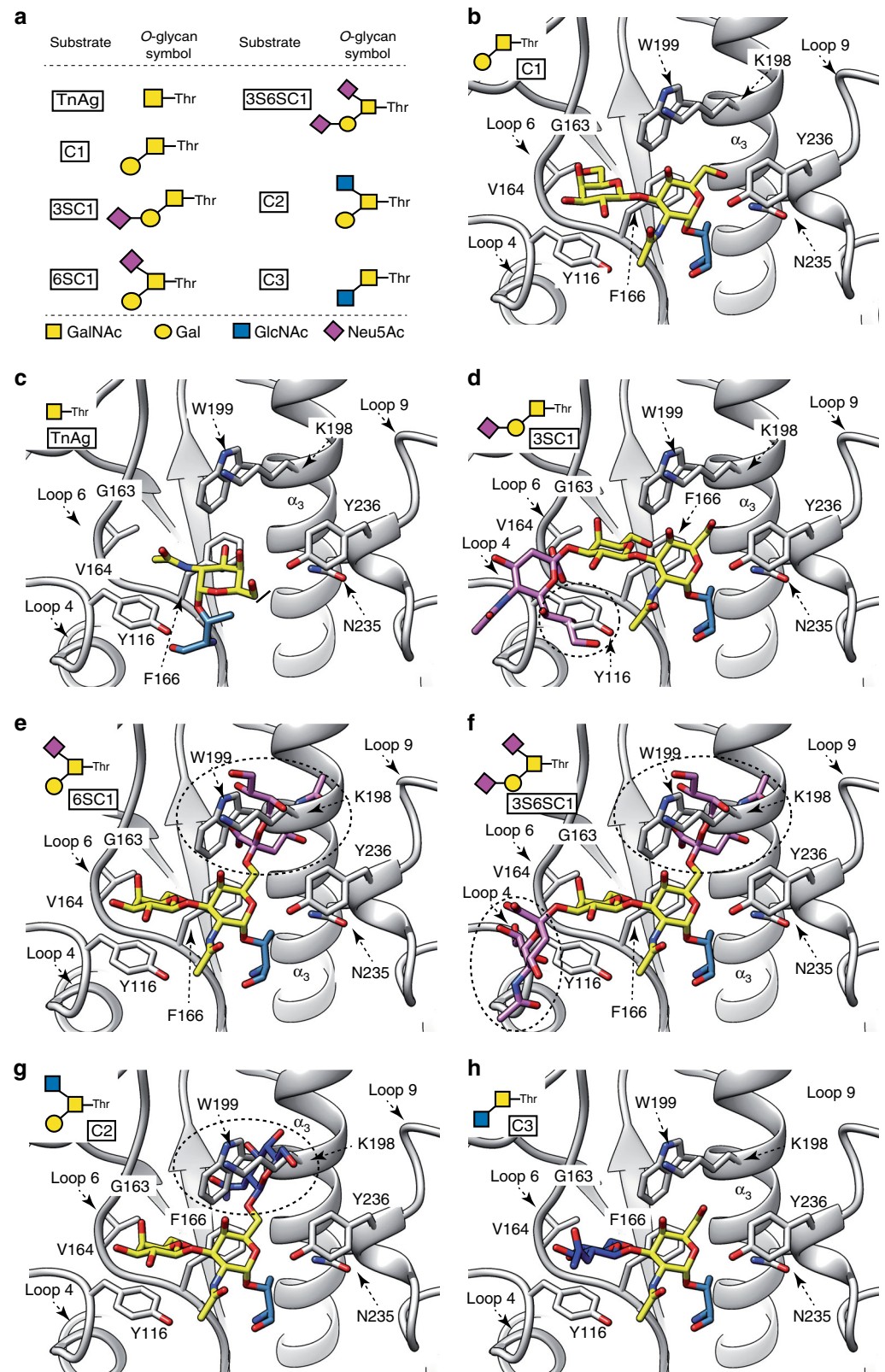

**Fig. 4 Structural basis of OgpA specificity for *O*-glycopeptides. a** *O*-glycopeptides used to measure OgpA activity by reverse-phase HPLC. **b** X-ray crystal structure of OgpA_WT-GD-PRO with C1 glycan. Docking calculations of OgpA with different *O*-glycans, including **c** Tn, **d** 3SC1, **e** 6SC1, **f** 3S6SC1, **g** C2 and **h** C3. The predicted clashes are shown as dotted circles.

**Table 1 Hydrolytic activity of OgpA determined by reverse-phase HPLC analysis.**

| O-glycopeptide | OgpA substrate | Initial turnover (fmol min$^{-1}$ unit$^{-1}$) |
|---|---|---|
| MUC1 | No | n.d. |
| TnAg | No | n.d. |
| C1 | Yes | 30.6 |
| 3SC1 | Yes | 0.18 |
| 6SC1 | No | n.d. |
| 3S6SC1 | No | n.d. |
| C2 | No | n.d. |
| C3 | Yes | 0.2 |

Neu5Ac-$\alpha$2,3-Gal-$\beta$1,3-(Neu5Ac-$\alpha$2,6)-GalNAc (3S6SC1), induce major steric repulsions with the long $\alpha_3$ helix of OgpA, as observed in C2 (Fig. 4e–g). As a consequence, the enzymatic activity is markedly reduced (3SC1) or completely abolished (6SC1, 3S6SC1, and C2; Table 1). Finally, slight shape changes in the second residue, as occurs in the C3 ligand, slows down activity 153-fold due to the proximity with the two mentioned connecting loops, more specifically, with residues Y116 an F166 (Fig. 4h). In this case, the only differences compared to C1 are the arrangement of the 4-OH group that converts galactose to glucose and the acetylation of the 2-position of the GlcNAc residue (Fig. 4h). Taken together, the combination of enzymatic activity measurements and molecular docking calculations, strongly support that OgpA is highly specific with respect of its O-glycan substrate, with the second Gal residue of C1 playing a prominent role.

## Discussion

To date, X-ray crystal structures of four enzymes with endo-peptidase activity against O-glycopeptides have been reported: (i) BT4244 from *B. thetaiotaomicron*[71], (ii) IMPa from *Pseudomonas aeruginosa*[71], (iii) ZmpB from *Clostridium perfringens* (strain ATCC 13124)[71], and (iv) StcE from enterohemorrhagic *Escherichia coli*[72]. The BT4244 and ZmpB enzymes belong to the M60 family, while StcE and IMPa belong to the M66 and M88 families, respectively. Members of the M60, M66, and M88 families belong to the MA clan. They share the common catalytic mechanism and the conserved metal-binding motif HEXXH described for OgpA (Supplementary Fig. 3). Members of the M60 and M88 families coordinate the Zn$^{2+}$ atom with two histidine residues and one aspartic acid, while members of the M66 family do not show the conserved methionine residue. Each of these enzymes shows a particular O-glycan specificity, different from that observed for OgpA (Supplementary Table 2). In vitro experiments on defined synthetic O-glycopeptides (TnAg, C1, 6SC1, and 3SC1) revealed that IMPa can hydrolyze TnAg, C1, 6SC1, and 3SC1, while BT4244 hydrolyzes TnAg, and ZmpB exclusively hydrolyzes 6SC1[71]. StcE cleaves densely O-glycosylated mucin proteins in the following residue to an O-glycosylated serine or threonine and shows activity on elaborated C1 and C2 structures[73]. The overall structures of BT4244, ZmpB, IMPa, and StcE are quite different compared to OgpA, and show additional domains. However, they all share some common structural elements that build the catalytic and the substrate-binding sites (Fig. 5a–d). All five enzymes show a central $\alpha$-helix ($\alpha_3$ in OgpA) and at least five-stranded $\beta$-sheet with different topology than OgpA. The nature and structural arrangement of the loops surrounding the O-glycan is also unique to each enzyme (Fig. 5a–d). X-ray crystal structures of BT4244, ZmpB, and IMPa in complex with an O-glycopeptide product revealed that the orientation of the glycan at the binding site also differs from that observed in OgpA. Specifically, in the OgpA-GD-PRO crystal structure, the O-glycan is found above $\alpha_3$ helix making interactions with residues of loops 4, 6, and 9, while in BT4244, ZmpB, and IMPa structures the equivalent loop 6 is shorter than in OgpA; also loops 4 and 6 are far from the O-glycan-binding site. This suggests that the loops that form the recognition subsites are different between the metalloprotease families. Moreover, the length of the $\alpha_3$ helix also differs markedly between these five enzymes (Fig. 5e). OgpA and BT4244 show a longer $\alpha_3$ helix than ZmpB, IMPa, and StcE. Molecular docking calculations clearly indicate that the OgpA $\alpha_3$ helix blocks the entrance of O-glycopeptides with substitutions in the O6 group of the first GalNAc residue (Fig. 4). Interestingly, BT4244 is unable to hydrolyze 2,6 sialylated O-glycopeptides, most likely due to the long $\alpha_3$ helix that blocks the access of the extra carbohydrate moiety at this position. Supporting this notion, ZmpB, IMPa, and StcE which shows a short $\alpha_3$, can hydrolyze 2,6 sialylated O-glycopeptides. In summary, the $\alpha_3$ helix is a common structural element among the metalloproteases of the gluzincins and metzincins families involved in the generation of the Zn$^{2+}$ binding motif, but also plays a key structural role in the selection of the O-glycan substrate in O-glycopeptidases[63].

The active site of metalloproteases has been defined in subsites that interact with the substrate side chains of the N-terminal region with respect of the scissile bond, non-primed side (P1, P2, P3, etc.), and the C-terminal region of the peptide to be cleaved, primed side (P1', P2', P3', etc.)[74,75]. The subsites of the metalloproteases that interact with the N-terminal region are named S1, S2, S3, and those that interact with the C-terminal region S1', S2', and S3'. This peptidase-substrate recognition paradigm has been updated for O-glycopeptidases[71], including subsites of the enzyme that specifically recognize the O-glycan moiety that has been termed G-sites. OgpA appears to show no preference for a specific amino acid sequence on the O-glycopeptide substrate. Indeed, a recent publication using OgpA mediated-digestion to map more than 3,000 O-glycosylation sites from different tissue and cell samples[76], specifically addressed this issue and found no evidence of sequence biases or digestion at sites lacking O-glycans. Previously known O-glycosylation sites were reliably detected regardless off sequence context further validating the method.

The crystal structure of OgpA$_{H205A/E206A}$-GD-SUB shows that protein-peptide interaction is primarily mediated by hydrogen bonds between the Pro-P3, Arg-P2, Pro-P1, and Ser-P1' backbone and the residues of OgpA that form the corresponding subsites (Fig. 6). The side chains of these amino acid residues also interact with OgpA by hydrophobic interactions. Mucins, the potential target for OgpA in vivo, consist of a long, densely O-glycosylated domain with sequences rich in Pro, Thr, and Ser, often characterized by tandem repeats. His-P3', Ser-P4, Tyr-P5, pro-P6, and Arg-P7 that form the O-glycopeptide are exposed to the solvent and do not interact with the protein. This indicates that the side chain nature of these residues is not a key element for OgpA activity. In that sense, the structural comparison of the four structures obtained for OgpA indicates that the substrate-binding site is essentially preformed, which supports a conformational selection mechanism for the recognition of the O-glycopeptide[77]. Finally, in the OgpA$_{H205A/E206A}$-GD-SUB, Thr-S1' subsite is wide and Thr-P1' barely interacts with OgpA. However, the Gal and GalNAc moieties of the O-glycan attached to this residue closely interacts with the enzyme's subsites G1' and G2' via hydrogen bonding, hydrophobic and CH–$\pi$ interactions, respectively. This suggests that the interaction between the OgpA G-sites and the O-glycan are key structural determinants for the recognition mechanism and the reaction to take place.

Both N- and O-linked glycosylations are considered critical quality attributes for therapeutic proteins and require careful

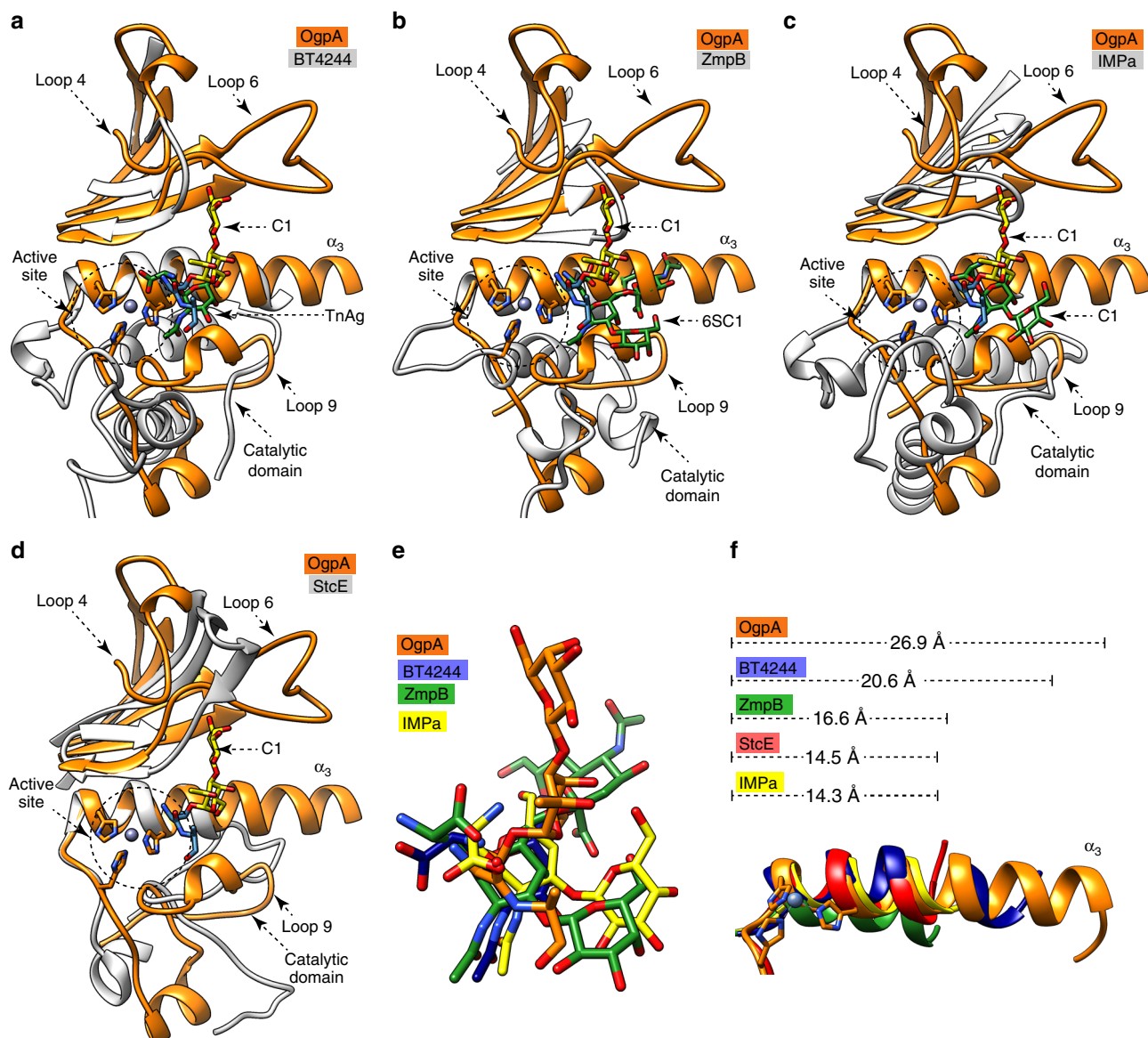

**Fig. 5 Structural basis of *O*-glycopeptide processing peptidases.** Structural comparison of OgpA and **a** BT4244 from *B. thetaiotaomicron* (PDB code 5KD8), **b** ZmpB from *C. perfringens* (strain ATCC 13124) (PDB code 5KDU) **c** IMPa from *P. aeruginosa* (PDB code 5KDX), and **d** StcE from the enterohemorrhagic *E. coli* (PDB code 3UJZ). The α-helices, β-strands and loops that interact with the *O*-glycans are represented in orange for OgpA, and in gray for the other enzymes. The *O*-glycopeptide product found in the crystal structure of OgpA is colored in yellow and blue. The *O*-glycopeptide product found in the crystal structures of BT4244, ZmpB, IMPa and StcE, is colored in green. **e** Superposition of the *O*-glycan found in the X-ray crystal structure of OgpA (orange), BT4244 (blue), ZmpB (green) and IMPa (yellow) **f** The α-helix length in each OgpA (orange), BT4244 (blue), ZmpB (green) and IMPa (yellow)and StcE (red) *O*-glycan endopeptidases complexes. .

analytical characterization during development and manufacturing[78,79]. Traditional sample preparation procedures, including enzymatic digestion with peptidases such as trypsin or LysC, have limitations in the efficient digestion of often densely packed *O*-glycosylated regions. The *O*-glycopeptidase OgpA has shown promise as a tool to map *O*-glycosylation sites since it generates a digestion pattern only where *O*-glycans are present. The enzymatic activity of OgpA combined with detailed analytical workflows, such as liquid chromatography coupled to mass spectrometry, have enabled detailed mapping of occupied *O*-glycosylation sites into therapeutic proteins such as human coagulation Factor V[80]. Specifically, using an OgpA-based analytical workflow, 18 *O*-glycosites were discovered in the B domain of Factor V compared to the 10 sites previously identified[81].

The OgpA enzyme also has the potential for further investigation into glycomics, such as the discovery of *O*-glycosylation sites in complex biological samples, including viruses or cancerous tissues. An OgpA-based chemoenzymatic analytical workflow has been developed for the extraction of *O*-linked glycopeptides (EXoO) for core 1 type structures. It uses the N-terminal proteolytic activity of OgpA to specifically release *O*-glycosylated peptides from a captured tryptic digest mixture[57,82]. Using mass spectrometry, over 3000 *O*-glycosites and their glycans could be identified in complex samples, such as kidney tissues, T-cells, and serum. Interestingly, this approach revealed differences in *O*-glycosylation profiles comparing healthy and tumor kidney tissues, indicating a possible application of the method for diagnostic or therapeutic use[57]. A similar workflow has been used to

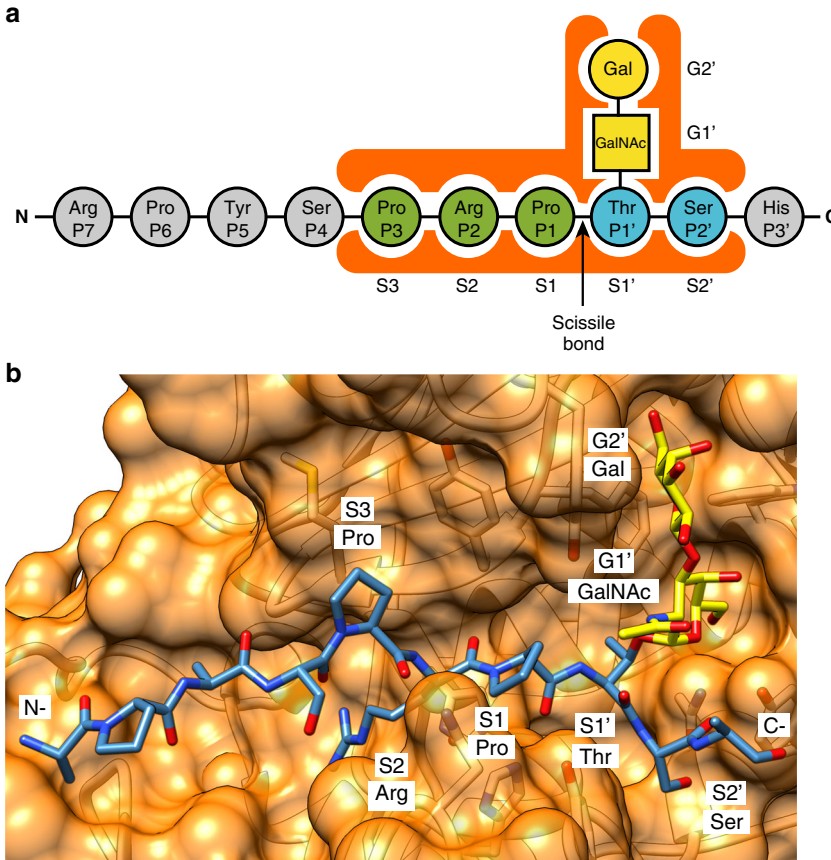

**Fig. 6 Subsite nomenclature for OgpA. a** The proposed subsite nomenclature for OgpA substrate recognition. Amino acids that are solvent exposed and not interacting with OgpA are colored in gray; amino acids of the N-terminal with respect of the scissile bond that interact with OgpA subsites are colored in green; amino acids of the C-terminal with respect of the scissile bond that interact with OgpA substites are colored in blue. Gal and GalNac residues are colored in yellow. **b** Surface representation of the active site of the OgpA_{H205A/E206A}-GD-SUB crystal structure with annotated subsites involved in substrate recognition.

study O-glycans in recombinant Zika virus proteins where 81 O-glycopeptides could be identified from the ZIKV-NS1 protein[57]. In the latter methodology, the sialic acid residues are derivatized to enable mass spectrometric analysis of specific linkages. These modifications remove the charge of sialic acids and increase the enzymatic activity of OgpA towards sialylated glycoforms.

Due to their high structural complexity and chemical diversity, different groups of enzymes participate in the degradation of mucins. According to the currently accepted model, the structural processing of mucins is initiated by the action of peptidases that first cleave the non-glycosylated moieties of the protein; followed by a series of α-glycoside hydrolases, which mainly remove the outermost glycan residues, such as GalNAc, Gal, and Fuc[83]. The resulting oligosaccharides are converted to monosaccharides by the action of β-glycoside hydrolases, while additional peptidases can complete the hydrolysis of the mucin backbone[83]. Therefore, it is accepted that the degradation of mucins involves the cooperative action between different microbes that populate the human gut.

*A. muciniphila* can hydrolyze up to 85% of the mucin structures by using a combination of different enzymatic activities, including proteases, sulphatases and glycoside hydrolases. Glycoside hydrolases can work on the peripheral side of the mucin, i.e., α-D-galactosidase, α-L-fucosidase and *N*-acetyl-α-D-galactosaminidase, as well as in the core molecule, i.e., β-D-galactosidase, β-D-fucosidase, *N*-acetyl-β-D-glucosaminidase, *N*-acetyl-β-D-galactosaminidase, β-D-glucosidase, and β-D-mannosidase[83]. Several gut microbes have developed a complex enzymatic machinery that degrades mucin, such as the *B. thetaiotaomicron*,

which contains polysaccharide utilization loci (PULs) that encode highly specific carbohydrate-active enzymes, surface glycan-binding proteins and transporters[22–24]. As mentioned above, BT4244 is an O-glycopeptidase from *B. thetaiotaomicron* comprised in PUL78, which also contains two genes, BT4241 and BT4243, encoding a GH2 and a GH109 glycoside hydrolases, respectively[84]. This indicates that O-glycopeptidases can work together with other enzymes (e.g., glycoside hydrolases) to process mucins. Very recently, crystal structures of endo-acting GH16 O-glycanases from *B. caccae* and *A. muciniphila* were reported[85]. The authors propose that these periplasmic O-glycanases initiate the degradation of O-glycans in mucins, followed by the action of exo-acting glycoside hydrolases before peptidases and O-glycopeptidases trim the remaining mucin backbone. Although there is no experimental evidence of Sus like systems in Verricomicrobium phyla[43], careful inspection of the protein-encoding genes in the genome of *A. muciniphila* ATCC BAA-835 strain showed that the *ogpA* gene is close to (i) a putative glycoside hydrolase of the GH95 family (Amuc_1120); this family is composed of enzymes with reported α-L-fucosidase, α-1,2-L-fucosidase and α-L-galactosidase activities, and (ii) a predicted sulphatase (Amuc_1118) (Supplementary Fig. 10, Supplementary Table 3). Both enzymes could also be part of the mucin degradation system. Recent studies showed that the expression of Amuc_1120 was found to be significantly upregulated under mucin *versus* glucose conditions[86]. However, neither OgpA nor Amuc_1118, a putative sulfatase, expression showed upregulation under the growth conditions studied[86]. These data do not exclude

the possibility that OgpA and Amuc_1118 could be involved in a common mucin degradation mechanism. These genes are also conserved in other *A. muciniphila sp.*, including *A. muciniphila CAG:154*, *A. muciniphila CAG:344 WG* and *A. muciniphila KLE1798*, (Supplementary Fig. 10), suggesting that these enzymes could collaborate together in the specific degradation of mucins.

In summary, we determine the molecular mechanism by which OgpA specifically recognizes and processes *O*-glycopeptides at the molecular level of detail. We provide the structure of the unliganded form of the enzyme and two structural snapshots of the reaction catalyzed by OgpA, the substrate-enzyme and the product-enzyme complexes. Both snapshots provide insights into the *O*-glycopeptide substrate binding and product release mechanisms. Based on these data, we identify other genes, neighboring *ogpA* gene, that encodes proteins potentially involved in mucin degradation such as glycoside hydrolases and sulfatases, suggesting that these enzymes could work together with others in a common mechanism. Although in recent years, substantial effort has been directed towards studying the enzymatic machinery of *Bacteroides*, further work is needed to understand the organization and mechanisms of mucin-degrading enzymes in *A. muchiniphila*, a key beneficial bacterium in the regulation of the gut barrier function and other physiological and homeostatic functions.

Finally, the discovery of OgpA has enabled new analytical workflows for the study of *O*-glycosylated proteins with applications in fundamental glycomics research and for therapeutic proteins in the biopharmaceutical industry. The increasing complexity of therapeutic proteins in clinical development further emphasizes the need for new sample preparation strategies to facilitate the characterization of *O*-glycosylation as a critical quality attribute. By enabling the mapping of *O*-glycosylation patterns, we believe that OgpA will certainly impact the development of future biologics and provide patients with better and safer therapies.

## Methods

**OgpA$_{WT}$ and OgpA$_{H205A/E206A}$ variant.** OgpA$_{WT}$ (OpeRATOR, Genovis) and OgpA$_{H205A/E206A}$ (GlycOCATCH, Genovis) were recombinantly expressed in *E. coli* and provided by Genovis. Lyophilized OgpA$_{WT}$ and OgpA$_{H205A/E206A}$ were dissolved in 20 mM Tris-HCl pH 7.6 and loaded into a Superdex 200 10/300 column (24 ml; GE Healthcare), equilibrated in 20 mM Tris-HCl pH 7.6, respectively (Supplementary Fig. 1). The eluted proteins were concentrated at 10 mg mL$^{-1}$ using an Amicon Ultra-4 centrifugal filter unit (Millipore) with a molecular cutoff of 10 kDa at 6500 × g, and aliquots stored at −80 °C.

**OgpA crystallization and data collection.** OgpA$_{WT}$ was crystallized in two crystal forms, referred thereafter as OgpA$_{WT1}$ and OgpA$_{WT2}$. OgpA$_{WT1}$ was crystallized by mixing 0.25 μL of a protein solution at 10 mg ml$^{-1}$ in 20 mM Tris-HCl pH 7.6, with 0.25 μL of 100 mM sodium HEPES/MOPS pH 7.5, 100 mM carboxylic acids mixture (sodium formate, ammonium acetate, sodium citrate tribasic dihydrate, sodium potassium tartrate tetrahydrate, and sodium oxamate) and 50% (w/v) of precipitant mix based on 40% (w/v) PEG 500 MME and 20% (w/v) PEG 20,000 (Morpheus® protein crystallization screen). The crystals grew in 1 day. The crystals were transferred to the crystallization solution and frozen under liquid nitrogen. The second crystal form OgpA$_{WT2}$ was obtained by mixing 0.25 μL of a protein solution at 10 mg ml$^{-1}$ in 20 mM Tris-HCl pH 7.6 with 0.25 μL of 100 mM sodium imidazole/MES monohydrate pH 6.5, 100 mM carboxylic acids mixture (sodium formate, ammonium acetate, sodium citrate tribasic dihydrate, sodium potassium tartrate tetrahydrate, and sodium oxamate) and 50% (w/v) of precipitant mix based on 25% (w/v) MPD, 25% (w/v) PEG 1000 and 25% (w/v) PEG 3350 (Morpheus® protein crystallization screen). The crystals grew in 1 day. The crystals were transferred to a cryo-protectant solution containing 10% ethylene glycol and frozen under liquid nitrogen. Complete X-ray diffraction datasets for both crystal forms were collected at beamline I24 (Diamond Light source, Oxfordshire, UK). OgpA$_{WT1}$ crystal crystallized in the tetragonal space group *P* 4$_1$ 2$_1$ 2 with one molecule in the asymmetric unit and diffracted to a maximum resolution of 1.9 Å (Supplementary Table 1). OgpA$_{WT2}$, crystallized in the orthorhombic space group *P* 2$_1$ 2$_1$ 2$_1$ with one molecule in the asymmetric unit and diffracted to a maximum resolution of 1.6 Å (Supplementary Table 1). OgpA$_{H205A/E206A}$-GD-SUB complex was crystallized by mixing 0.25 μL of a protein solution of OgpA$_{H205A/E206A}$ at 10 mg ml$^{-1}$ in 20 mM Tris-HCl pH 7.6 and 2.5 mM GD, with 0.25 μL of 100 mM MIB pH 5.0 and 25% (w/v) PEG 1,500. The crystals grew in 1 day. The crystals

were transferred to a cryo-protectant solution containing 25% ethylene glycol and 2.5 mM GD, and frozen under liquid nitrogen. A complete X-ray diffraction dataset was collected at beamline BL13-XALOC (ALBA, Cerdanyola del Valles, Spain). OgpA$_{H205A/E206A}$-GD-SUB complex crystallized in the tetragonal space group *I* 4 with one molecule in the asymmetric unit and diffracted to a maximum resolution of 2.16 Å (Supplementary Table 1). OgpA$_{WT}$-GD-PRO complex was crystallized by mixing 0.25 μL of a protein solution of OgpA$_{WT}$ at 10 mg ml$^{-1}$ in 20 mM Tris-HCl pH 7.6 and 2.5 mM GD, with 0.25 μL of μL of 100 mM sodium HEPES/MOPS pH 7.5, 100 mM carboxylic acids mixture (sodium formate, ammonium acetate, sodium citrate tribasic dihydrate, sodium potassium tartrate tetrahydrate and sodium oxamate) and 50% (w/v) of precipitant mix based on 40% (w/v) PEG 500 MME and 20% (w/v) PEG 20,000 (Morpheus® protein crystallization screen). The crystals grew in 1 day. The crystals were transferred to the crystallization solution and frozen under liquid nitrogen. A complete X-ray diffraction dataset was collected at beamline I03 (Diamond Light source, Oxfordshire, UK). The OgpA$_{WT}$-GD-PRO complex crystallized in the tetragonal space group *P* 4$_1$ 2$_1$ 2 with one molecule in the asymmetric unit and diffracted to a maximum resolution of 2.34 Å (Supplementary Table 1). All datasets were integrated and scaled with XDS following standard procedures[87].

**Structures determination and refinement.** The crystal structure of full length OgpA in its unliganded form was solved using zinc single-wavelength anomalous dispersion (Zn-SAD)[88]. Anomalous data of an OgpA$_{WT2}$ crystal were collected at the theoretical 9.6586 keV absorption edge of Zn (9.669 keV eV−1.28228 Å; OgpA$_{WT2-SAD}$). The data were integrated and scaled with XDS[86]. The substructure determination located two Zn atoms in the asymmetric unit (CC = 12.6, CC(weak) = 7.6 and CFOM = 20.2). Experimental phases were determined using the Big EP pipeline[89]. The partial model produced with CRANK2 was subsequently used for initial cycles of model building, density modifications, and refinement for Buccaneer and the CCP4 suite[90–92]. The structure determination of OgpA$_{WT1}$, OgpA$_{WT2}$, OgpA$_{H205A/E206A}$-GD-SUB and OgpA$_{WT}$-GD-PRO were carried out by molecular replacement using Phaser and the PHENIX suite and the OgpA$_{WT-SAD}$ structure as initial model[93,94]. Followed by cycles of manual rebuilding and refinement using Coot and phenix.refine, respectively[95,96]. The structures were validated by Mol-Probity[97]. Data collection and refinement statistics are presented in Supplementary Table 1. The atomic coordinates and structure factors have been deposited with the Protein Data Bank, accession codes 6Z2D, 6Z2O, 6Z2P, and 6Z2Q. Molecular graphics and structural analyses were performed with the UCSF Chimera package[98].

**Structural analysis and sequence alignment.** Structure-based sequence alignment analysis were performed using Chimera[98]. Protein pocket volume was calculated using HOLLOW[99]. Z-score values were produced by using DALI[100]. Domain interface analysis was performed using PISA[101]. Conserved and similar residues were labeled using BoxShade server (http://embnet.vital-it.ch/software/BOX_form.html).

**Synthesis of *O*-glycosylated peptides.** All *O*-glycopeptides used in this study are described in Fig. 4. The 3SC1 glycopeptide was purchased from Sussex Research. The 3SC1 glycopeptide at 400 μM in 25 μl of 50 mM Tris-HCl pH 7.0 was treated with (i) 20 units of α-2,3-sialidase (SialEXO23; Genovis AB, Lund), (ii) 20 units of α-2,3-sialidase (SialEXO23; Genovis AB, Lund) and 20 units of β-galactosidase (GalactEXO; Genovis AB, Lund) or (iii) 20 units of α-2,3-sialidase (SialEXO23) and 20 units of *O*-glycosidase (OglyZOR; Genovis AB, Lund) at 37 °C overnight, to generate peptides C1, Tn and the unmodified control peptide, respectively. The C2 glycopeptide was synthesized by incubating the glycopeptide C1 at 37 °C for 2 h, under the following conditions: 400 μM glycopeptide, 10 mM UDP-GlcNAc and 2 μg recombinant human β-1,3-galactosyl-*O*-glycosyl-glycoprotein β-1,6-N-acetylglucosaminyltransferase (GCNT1; R&D Systems) in 50 μl 100 mM MES, pH 6.0, supplemented with 5 mM CaCl$_2$ and 10 mM DTT. For the synthesis of glycopeptide C3, glycopeptide Tn (400 μM) was incubated at 37 °C for 2 h, with 10 mM UDP-GlcNAc and 2 μg recombinant human acetylgalactosaminyl-*O*-glycosyl-glycoprotein β-1,3-N-acetylglucosaminyltransferase (B3GNT6; R&D Systems) in 50 μl 25 mM Tris-HCl, pH 7.5, 150 mM NaCl, 10 mM MnCl$_2$, 5 mM CaCl$_2$ and 20% DMSO. Glycopeptides 6SC1 and the di-sialylated 3S6SC1 were synthesized by incubating 400 μM C1 and 3SC1, respectively, with 2 μg α-N-acetylgalactosaminyl α-2,6-sialyltransferase 2 (ST6GALNAC2; R&D Systems) and 10 mM CMP-Neu5Ac in 50 μl 25 mM Tris-HCl, pH 7.5, 10 mM MnCl$_2$. The enzymes were removed from the reactions by passing the reaction mixture through a 10 kDa MWCO filter unit (PALL NanoSep). All glycopeptides were cleaned-up using Pierce C18 tips (Thermo Scientific), evaporated to dryness in a vacuum centrifuge and dissolved in 20 mM Tris-HCl, pH 7.0.

**OgpA activity assays.** The glycopeptides were incubated with the indicated amounts of OgpA and the reaction was stopped at different time points by diluting the reaction mixture in 0.1% FA. Turnover was analyzed by reverse-phase HPLC on a Vanquish Duo UHPLC system equipped with an MSPac DS-10 desalting cartridge (both Thermo Fisher). The glycopeptides were separated by a 5 min. gradient from 1.8 to 12% ACN in 5 mM sodium phosphate pH 8.0 at 50 °C and detected by fluorescence of the 5-FAM fluorophore (excitation :494 nm, emission:

521 nm). Quantification of product and substrate peaks allows for determination of product formation according to the following formula (Eq. (1)):

$$n_{\text{product}} = \frac{peak\ area_{\text{product}}}{peak\ area_{\text{educt}} + peak\ area_{\text{product}}} \times n_{\text{total}} \qquad (1)$$

where $n_{\text{product}}$ refers to the amount of product formed (i.e., digested peptide), $peak\ area_{\text{product/substrate}}$ to the measured area under the curve for the product and substrate respectively and $n_{\text{total}}$ is the total amount of peptide in the reaction. From this data, the initial turnover rate of each reaction could be determined by linear regression.

**Molecular docking calculations.** The Tn, 3SC1, 6SC1, 3S6SC1, C2, and C3 O-glycans were modeled using GLYCAM-Web website (Complex Carbohydrate Research Center, University of Georgia, Athens, GA; http://www.glycam.com). Ligand docking was performed using AutoDock Vina employing standard parameters and visualized using USCF Chimera[98,102]. The full protein-ligand interaction profiles OgpA$_{\text{H205A/E206A}}$-GD-SUB (PDB code 6Z2P) and OgpA$_{\text{WT}}$-GD-PRO (PDB code 6Z2Q) were generated using LIGPLOT[103].

**Reporting summary.** Further information on research design is available in the Nature Research Reporting Summary linked to this article.

## Data availability

The atomic coordinates and structure factors have been deposited with the Protein Data Bank (PDB), accession codes 6Z2D (OgpA$_{\text{WT1}}$), 6Z2O (OgpA$_{\text{WT2}}$), 6Z2P (OgpA$_{\text{H205A/E206A}}$-GD-SUB), and 6Z2Q (OgpA$_{\text{WT}}$-GD-PRO). The following PDB accession codes have been used for analysis in this manuscript: 3G42, 1ATL, 1ND1, 5JIP, 6H7W, 6BYI, 5KD8, 5KDU, 5KDX, 3UJZ. The protein sequences of OgpA from *Akkermansia muciniphila* (strain ATCC BAA-835/Muc), BN502_01341 from *A. muciniphila CAG:154*, DDX86_00850 from *Akkermansia* sp., BN616_02307 from *Akkermansia* sp. CAG:344, DF185_09660 from *Marinifilum breve*, BN783_01361 from *Odoribacter* sp. CAG:788, SAMN04488055_0630 from *Chitinophaga niabensis*, ECE50_27605 from *Chitinophaga* sp. Mgbs1, SAMN05660461_5684 from *Chitinophaga ginsengisegetis*, ATE49_15630 from *Elizabethkingia miricola*, BD94_1796 from *Elizabethkingia anophelis NUHP1*, IQ37_10980 from *Chryseobacterium piperi*, P278_02110 from *Zhouia amylolytica*, W5A_07577 from *Imtechella halotolerans* are obtained from UniProt database. All other data that support the findings of this study are available from the corresponding authors on reasonable request.

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

## Acknowledgements

This work was supported by the MINECO/FEDER EU contracts BFU2016-77427-C2-2-R, BFU2017-92223-EXP and Severo Ochoa Excellence Accreditation SEV-2016-0644; the Basque Government contract KK-2019/00076 and NIH R01AI149297 (to M.E.G.). This project has received funding from the European Union's Horizon 2020 research and innovation programme under the Marie Skłodowska-Curie grant agreement No. 844905 (B.T) and the Basque Government (I.A.). We acknowledge Diamond Light Source (proposals mx20113), ALBA synchrotron beamline BL13-XALOC (mx2018093013), and iNEXT (proposals 1618/2538) for providing access to synchrotron radiation facilities. We gratefully acknowledge all members of the Structural Glycobiology Lab (CIC bioGUNE, Spain) for valuable scientific discussions. We would like to thank Fredrik Leo, Rolf Lood, and Linda Andersson at Genovis for providing material. Glycodrososcin was kindly provided by Prof. Mattias Collin at Lund University.

## Author contributions

B.T., A.N., J.S., and M.E.G., conceived the project. B.T., A.N., and I.A., performed the experiments. B.T., A.N., I.A., J.S., and M.E.G., analyzed the results. B.T., A.N., I.A., J.S., and M.E.G., wrote the paper.

## Competing interests

J.S. and A.N. are employees of Genovis A.B., and A.N. hold shares in the company. B.T., I.A., and M.E.G. declare no competing interests.

## Additional information

**Peer review information** *Nature Communications* thanks Alisdair Boraston and other, anonymous, reviewers for their contributions to the peer review reports. Peer review reports are available.

