## [Peer Review File · Nature Communications]

REVIEWER COMMENTS

Reviewer #1 (Remarks to the Author):

This manuscript describes the specificity, and structural basis thereof, of the OgpA "O-glycan protease" from *Akkermansia muciniphila*. In my opinion, this is a very exciting area of research that is expanding in its relevance and interest base. I found this particular manuscript to be very strong and I feel it would ultimately be appropriate for publication. In particular, being able to capture the structure of mutationally inactivated OgpA with an intact glycopeptide substrate that spans the catalytic machinery is a wonderful advancement. My comments are generally quite minor, and in some cases picky, but I wish the authors to consider them carefully. First, however, the revised manuscript has to be carefully copyedited for grammar and language. I started to list corrections but got a bit overwhelmed and abandoned this effort.

1. The authors have chosen to use the term "O-glycan protease" in the title and throughout the manuscript. Having personally worked extensively on enzymes that hydrolyze the bonds in glycans as well as having done some work on peptidases, I find the terminology chosen by the authors to read as internally contradictory - it implies proteolysis of glycans, which of course is impossible. I would suggest changing the terminology. "Glycopeptidase" and "glycoprotease," which indicate either glycopeptide or glycoprotein substrates of a peptidase/protease, have gained traction. O-glycopeptidase or O-glycoprotease might make a suitable alternative that better aligns with what I feel the authors were trying to get across.

2. Throughout the manuscript, the authors appear to have conflated the Peptidase_M60 family (defined by Pfam 13402) with MEROPS M60. IMPa, BT4244, ZmpB, and SsIE (aka YghJ) are all in Peptidase_M60 (PF13402). However, a close examination of the Noach et al. article and the MEROPS database will reveal that IMPa is in M88, BT4244 and ZmpB are in M60, and SsIE (aka YghJ) is in M98. Peptidase_M60 is an extremely large family, more of a superfamily, where at the extremes of the family the sequences can almost appear unrelated.

3. The authors state that OgpA falls into MEROPS family M11. I think that this should be better supported in the manuscript. I looked in MEROPS and there are no *Akkermansia* sp. entries in that family. Furthermore, the homologs that the authors identify in supplementary alignment also are not in M11. I BLASTED the OgpA sequence and could not identify any known members of M11 in the hit list. So, I am not sure what the basis of the authors' classification as M11 is. Is it possible that this should be a new MEROPS family? Anyways, clarification is warranted. I think this point, and the previous point, which regard carefully and appropriately placing these proteins in the context of "family relatedness", are important as this field moves forward.

4. In the discussion, the authors compare the activity of OgpA to BT4244 and state that BT4244 is active on C1 (i.e. a glycan bearing the Core 1 or T-antigen). This is not supported by the data in the Noach et al article, which shows activity on the Tn-antigen but not the T-antigen or more elaborate glycans.

5. Figure 2 is extraordinarily busy with so many labels. It is really hard to extract the salient information. I think this figure needs to be simplified, which could begin with not labeling the loops in every panel - that they are colored the same in every panel is clear. Furthermore, it lacked a visual presentation of the more important features I was looking for, i.e. the specific interactions between substrate and enzyme. The sidechains are shown, but this doesn't reveal which ones might hydrogen bond (which is typically shown with dashes), while the flattened perspective of the image rendering made it a bit difficult to see how the aromatic amino acids played into the interaction. Perhaps the authors might find it useful to remove one or two of panels e, f, g and replace it with a 2D schematic?

6. I found Figure 3 to be largely redundant with Figure 2, as the product is simply a fragment of the GD substrate. I found Supplementary Figure 4 to contain the relevant result regarding the product complex and, overall, to be much more useful to the story. Figure 3 and Supplementary Figure 4 could simply be swapped.

7. I like the concept of Figure 7 but, again, something about the way these panels were rendered really flattens out the perspective. This makes it difficult to quickly identify the predicted clashes, which is obviously key to the purpose of this analysis and display item. Perhaps restoring the depth cue and/or shadows would help with this? Or circling the site of the clash?

8. I am not sure how appropriate it is to cite a thesis (reference 84). I took a look at the relevant chapter in the thesis and do not necessarily doubt the veracity of the content, but it is not an officially peer-reviewed article (personally, I have never seen a thesis cited in a published article before). The authors may find this article - PMID: 28687644 - as a peer-reviewed and compelling argument for the likely combined roles of peptidases and GHs in mucin metabolism by *Akkermansia muciniphila*. Curiously, however, the gene encoding OgpA (Amuc_1119) does not appear to be upregulated during growth on mucin. In fact, it looks like none of the genes in that genomic region are substantially upregulated. Given that the authors set the stage for the thesis of their own manuscript by discussing *Akkermansia* and its role in mucin degradation, yet OgpA does not seem to be deployed by the microbe for this biological purpose, the authors might consider addressing this issue in the manuscript.

9. A less important issue is that I think the first section of the results can be shortened a bit. There is a lot of information in there that is not central to the story and even hardcore structural biologists would find unnecessary, e.g. detailed descriptions of secondary structure, protein dimensions, spacegroups, etc. All that information is either visually evident in the figures, can be derived from the stats tables, or will ultimately be available when the PDBs are public.

10. With all due respect to the authors, significant portions of this manuscript come across as an advertisement for a product. For example, the authors specifically refer to the OperATOR and GlycOCATCH products in the introduction, and 1-2 paragraphs in the discussion largely point out success stories of these products. I am not sure how appropriate this is, and I personally found it quite distracting from a very strong study that tells an otherwise interesting story. However, this is my opinion and not a catastrophic issue, so I will leave this to the discretion of the Editor.

Alisdair Boraston

Reviewer #2 (Remarks to the Author):

Trastoy et al have submitted a manuscript on the structure of a glycopeptide hydrolyzing protease from *Akkermansia muciniphila* including complexes with a glycopeptide substrate and a product derived thereof, respectively. This very interesting study also includes well comparison with structures of other related metalloproteases including such with known O-glycan specificity. The manuscript reads easily, although the figures can benefit from more labelling as detailed below. Moreover, it may be useful to make subheadings in the long Discussion. Some points to consider are listed below.

Major:

1. There seems to be claimed "essentially" no specificity on the peptide part for the O-glycan substrate. Can this be further explained? Can in fact "any sequence be accommodated" ? How will specificity on the peptide part be seen in the light of the single substrate sequence analysed in the present manuscript? How many different peptide sequences will there be with relevant O-glycan units in the niche habitat of this bacterium ?
2. When the authors say (page 6) they will use the WT2 form for the description, then why is WT1

shown in Figure 1?

3. On page 6, describing the topology the beta-4 strand is mentioned to be antiparallel, so when the peptide substrate is mentioned elsewhere to be parallel, is this then to beta-4 or to the other four strands of the beta-sheet ? (I assume the latter, but please clarify). One reason for clarifying can be that it would in fact be very helpful if this clear insertion of the substrate as a additional strand in the sheet can be illustrated on the relevant figures, maybe also the beta-strands or some of them can be numbered on relevant figures. The insertion into the beta-sheet is not the impression one at the moment can obtain from the current figure 2, but I don't know if that will be simple to illustrate by molecular graphics?

4. Can the authors help the readers by connecting M11, M12 and M60 families, perhaps even on page 7?

5. When it is mentioned that the M12 enzymes show no specificity against glycopeptides, does this mean that they are not able to cleave glycopeptides or that they can cleave these as well as non-glycosylated peptides, in other words these other enzymes just don't need the glycosylation in the substrates?

6. I assume that the full length GD was in the structure, but that density was not seen for the residues 1-3 and 14-19. One may read that it is a truncated form present in the structure, although admittedly the authors refer to "density".

7. Which kind of electrostatic interaction do the authors have in mind between the 2-acetamido group on GalNAc and the O2 of Gal? (page 9)

8. Which acetylation of the 6-position do the authors think of when comparing C1 and C3? (page 12)

9. When the authors state that the specificity differs for the various O-glycan recognizing proteases with regard to the O-glycan recognition. Does this then consider that there can be prominent and poor substrates for these enzymes as demonstrated in the present work in the case of OgpA? One may ask if other another enzyme also prefers C1? And if that is the case then perhaps also have very low activity on the other substrates listed in the text. Moreover, in the HPLC chromatograms in the Supplemental Information where the asterisks are marking the position of the main product, will there be a trace in other candidate substrates tested than the two where the product of the poor activity is shown in a "blow-up"?

10. Although belonging to the Materials, it would be good to have proper names for the various glycoside hydrolases applied to generate the substrates rather than just the company name for the enzymes.

11. In Figure 1 it is a bit unclear what is the idea in panel b? That the "OCATCH" can select and hence purify an O-glycan peptide? There are shown structures from very many different view-angles, may it be possible to label for example alpha helix 3 on several figures and other central elements to help the readers? This includes figures in the Supplementary Information.

12. On Supplementary Figure 2 can some structural elements be tagged? Secondly are the peptide SUB and PRO not seen on the part of the molecule? (panels c and d). Is it beta-strand 4 which is yellow on panel c? And why is this?

13. How long reaction time was used for samples subjected to HPLC analysis in Supplemental Figure 6.

Minor:

1. In the abstract correct to "...A. muciniphila uses to..."

2. In the abstract rephrase to "...that exclusively hydrolyzes the peptide bond...residues substituted with..." – If this is correctly understood by this Reviewer. (same point on page 5)

3. Page 3, correct to "...of fully processing a small..."

4. Page 3, correct to "...depolymerization of glycan structures..."

5. Page 3, correct two grammatical problems to "...contains polysaccharide utilization loci that encode highly..."

6. Correct to Gram (not gram, this is named after a Danish bacteriologist)

7. Page 5, correct to "...specific digestion patterns that..."

8. Page 8, correct to "...of an N-terminal catalytic..."

9. Page 17, correct to "..., the latter acting..."

10. D for carbohydrate stereo form should be in small cap

11. Page 18, correct to "... effort has been..."
12. In the METHODS please refer to the SI where relevant for example on the production of the recombinant protein. Can the yields be informed on the recombinant proteins?
13. Page 23, last line correct to phosphate
14. This Reviewer is in doubt what is meant by "...product and educt.." what is "educt"? (page 24)

Reviewer #3 (Remarks to the Author):

The significance of a well-balanced microbiome for the individuals' health status is well documented, but the molecular mechanisms that establish the mutualistic interactions between host and microbiome are far from being understood. Nevertheless, the current knowledge allows to state that the complementary benefit arises to a large part via carbohydrate metabolism. The manuscript by Trastoy et al. focuses at this domain and presents the X-ray crystal structure of a microbial enzyme O-glycan protease (OgpA) from a key beneficial species (*Akkermansia muciniphila*) involved in the control of the gut barrier thickness and function. OgpA can degrade host mucins, a group of heavily O-glycosylated proteins that are essential in barrier integrity. OgpA cuts the protein backbone in dependence of O-glycans and this study reveals the molecular basis for the enzymatic mechanism and specificity. Crystals were obtained for the wild-type apo enzyme and in complex with the product. A functionally inactive variant (OgpA-H205A/E206A) was used to solve the substrate ligated structure. The combined data together with in vitro enzymatic studies on the 8 most common O-glycan variants and molecular docking studies allowed the detailed description of the enzymatic mechanism and revealed the basis for the enzymes' specificity.

This is a well performed and comprehensive study, providing important novel data towards the understanding of the mutualistic host microbiome interactions and simultaneously fundamental mechanistic insight in an enzyme of utmost value in mucin analytics. In my eyes this study is ready for publication after a careful check with respect to grammatical issues. There are numerous mistakes in verb forms (plural/singular); Gram (Hans Christian Gram 1853–1938) is a proper name and must be capitalised. There are more minor spelling errors that need to be addressed.

Rita Gerardy-Schahn

Reviewer #1: Prof. Alisdair Boraston

This manuscript describes the specificity, and structural basis thereof, of the OgpA “O-glycan protease” from *Akkermansia muciniphila*. In my opinion, this is a very exciting area of research that is expanding in its relevance and interest base. I found this particular manuscript to be very strong and I feel it would ultimately be appropriate for publication. In particular, being able to capture the structure of mutationally inactivated OgpA with an intact glycopeptide substrate that spans the catalytic machinery is a wonderful advancement. My comments are generally quite minor, and in some cases picky, but I wish the authors to consider them carefully.

ANSWER: We very much appreciate that you consider our study very strong, and all the constructive comments/suggestions to improve the article.

First, however, the revised manuscript has to be carefully copyedited for grammar and language. I started to list corrections but got a bit overwhelmed and abandoned this effort.

ANSWER: Many thanks to point this to us. We have made a substantial effort to improve the grammar and readability of the entire manuscript.

1. The authors have chosen to use the term “O-glycan protease” in the title and throughout the manuscript. Having personally worked extensively on enzymes that hydrolyze the bonds in glycans as well as having done some work on peptidases, I find the terminology chosen by the authors to read as internally contradictory - it implies proteolysis of glycans, which of course is impossible. I would suggest changing the terminology. “Glycopeptidase” and “glycoprotease,” which indicate either glycopeptide or glycoprotein substrates of a peptidase/protease, have gained traction. O-glycopeptidase or O-glycoprotease might make a suitable alternative that better aligns with what I feel the authors were trying to get across.

ANSWER: We agree with the reviewer. We have replaced the term “O-glycan protease” by “O-glycopeptidase”.

2. Throughout the manuscript, the authors appear to have conflated the Peptidase_M60 family (defined by Pfam 13402) with MEROPS M60. IMPa, BT4244, ZmpB, and SsIE (aka YghJ) are all in Peptidase_M60 (PF13402). However, a close examination of the Noach et al. article and the MEROPS database will reveal that IMPa is in M88, BT4244 and ZmpB are in M60, and SsIE (aka YghJ) is in M98. Peptidase_M60 is an extremely large family, more of a superfamily, where at the extremes of the family the sequences can almost appear unrelated.

ANSWER: We agree with the reviewer. We have modified the text accordingly.

3. The authors state that OgpA falls into MEROPS family M11. I think that this should be better supported in the manuscript. I looked in MEROPS and there are no *Akkermansia* sp. entries in that family. Furthermore, the homologs that the authors identify in supplementary alignment also are not in M11. I BLASTED the OgpA sequence and could not identify any known members of M11 in the hit list. So, I am not sure what the basis of the authors’ classification as M11 is. Is it possible that this should be a new MEROPS family? Anyways, clarification is warranted. I think this point, and the

previous point, which regard carefully and appropriately placing these proteins in the context of “family relatedness”, are important as this field moves forward.

ANSWER: Many thanks to point this to us. We agree with the reviewer.

According to the search we have performed:

1. NCBI search - Conserved Domain: OgpA belongs to the *Peptidase_M11*.
2. NCBI search – Gene: AMUC_RS11990 hypothetical protein [*Akkermansia muciniphila* ATCC BAA-835]: OgpA belongs to the *Peptidase_M11*.

However, as the reviewer states:

3. MEROPS M11 search: There are no *Akkermansia* sp. entries in this MEROPS family.
4. NCBI search – BLAST: There are no known members of M11 in the hit list. The closest homologues belong to the family M12. Structurally, the closest homologues also belong to the family M12. However, it is worth noting that the family M11 is structurally uncharacterized.

In the new version of the manuscript, we have decided to incorporate a discussion on this issue:

“The OgpA catalytic domain is annotated as an M11 peptidase, also called the gametolysin peptidase M11 family, according to the NCBI database. However, there are no *Akkermansia* sp. entries for the M11 family in MEROPS, a database that uses a hierarchical, structure-based classification of peptidases.⁶⁴ Interestingly, amino acid sequence homology analysis using the MEROPS scan data set (MEROPS-MPRO) indicates that the OgpA metal-binding motif shares high sequence identity with peptidases classified in the M12 family. Both families, M11 and M12, share the common motif HEXXHXXXXXH, in which the three His residues are zinc ligands and the Glu residue has a catalytic function. Structurally, the closest homologues also belong to the M12 family. However, it is worth noting that the M11 family is not structurally characterized. The M11 and M12 families belong to the MA(M) subclass, which comprises a “metzincin” fold, with a methionine C-terminal to the Zn²⁺ atom.^{63,65} Members of M11 and M12 families are proenzymes that require activation by limited proteolysis.⁶³ There is no experimental evidence to support that OgpA is a proenzyme. Taken together, all the data suggests that OgpA could be consider as a member of a new family of peptidases, but more experimental data, i.e. the three-dimensional structure of a former M11 family member, and detailed analysis are needed.”

4. In the discussion, the authors compare the activity of OgpA to BT4244 and state that BT4244 is active on C1 (i.e. a glycan bearing the Core 1 or T-antigen). This is not supported by the data in the Noach et al article, which shows activity on the Tn-antigen but not the T-antigen or more elaborate glycans.

ANSWER: We agree with the reviewer. We have modified the text accordingly.

5. Figure 2 is extraordinarily busy with so many labels. It is really hard to extract the salient information. I think this figure needs to be simplified, which could begin with not labeling the loops in every panel – that they are colored the same in every panel is clear. Furthermore, it lacked a visual presentation of the more important features I was looking for, i.e. the specific interactions between substrate and enzyme. The sidechains are shown, but this doesn't reveal which ones might hydrogen bond (which is typically shown with dashes), while the flattened perspective of the image rendering

made it a bit difficult to see how the aromatic amino acids played into the interaction. Perhaps the authors might find it useful to remove one or two of panels e, f, g and replace it with a 2D schematic?

ANSWER: Many thanks for the suggestion. We have made an effort to improve the Figure 2. First, we have removed the labeling of the loops as suggested. Second, we have incorporated key interactions between the substrate and the enzyme, in the form of dashed lines. Finally, we have incorporated a 2D schematic Figure, showing the entire contacts between the substrate and product with the enzyme, as a new Supplementary Figure 5.

6. I found Figure 3 to be largely redundant with Figure 2, as the product is simply a fragment of the GD substrate. I found Supplementary Figure 4 to contain the relevant result regarding the product complex and, overall, to be much more useful to the story. Figure 3 and Supplementary Figure 4 could simply be swapped.

ANSWER: Many thanks for the suggestion. However, we believe that Figure 3 is also relevant for the reader, showing (i) the location of the product into the active site and (ii) the structural comparison between the substrate and product. For those reasons, we have decided to incorporate (i) the previous Supplementary Figure 4 (Supplementary Figure 6 in the new version of the manuscript), and (ii) the description of the catalytic mechanism of OgpA, into the Supplementary Information.

7. I like the concept of Figure 7 but, again, something about the way these panels were rendered really flattens out the perspective. This makes it difficult to quickly identify the predicted clashes, which is obviously key to the purpose of this analysis and display item. Perhaps restoring the depth cue and/or shadows would help with this? Or circling the site of the clash?

ANSWER: We agree with the reviewer. We have circled the clashes in the corresponding panels of Figure 4.

8. I am not sure how appropriate it is to cite a thesis (reference 84). I took a look at the relevant chapter in the thesis and do not necessarily doubt the veracity of the content, but it is not an officially peer-reviewed article (personally, I have never seen a thesis cited in a published article before).

ANSWER: We understand the reviewer's concern. However, the thesis (reference 84) has been previously cited in the following peer-reviewed articles:

1. Collado, M. C., Derrien, M., Isolauri, E., De Vos, W. M., and Salminen, S. (2007) Intestinal integrity and *Akkermansia muciniphila*, a mucin-degrading member of the intestinal microbiota present in infants, adults, and the elderly. *Appl. Environ. Microbiol.* 73, 7767–7770.

2. Geerlings, S., Kostopoulos, I., de Vos, W., and Belzer, C. (2018) *Akkermansia muciniphila* in the Human Gastrointestinal Tract: When, Where, and How? *Microorganisms*, 6, 75.

We prefer to cite the thesis in the manuscript.

The authors may find this article - PMID: 28687644 – as a peer-reviewed and compelling argument for the likely combined roles of peptidases and GHs in mucin metabolism by *Akkermansia muciniphila*.

Curiously, however, the gene encoding OgpA (Amuc_1119) does not appear to be upregulated during growth on mucin. In fact, it looks like none of the genes in that genomic region are substantially upregulated. Given that the authors set the stage for the thesis of their own manuscript by discussing Akkermansia and its role in mucin degradation, yet OgpA does not seem to be deployed by the microbe for this biological purpose, the authors might consider addressing this issue in the manuscript.

ANSWER: Many thanks to point this to us. Very useful information. We have incorporated this information in the Discussion section.

“Recent studies showed that the expression of Amuc_1120 was found to be significantly upregulated under mucin *versus* glucose conditions.⁸⁷ However, neither OgpA nor Amuc_1118, a putative sulfatase, expression showed upregulation under the growth conditions studied.⁸⁷ These data do not exclude the possibility that OgpA and Amuc_1118 could be involved in a common mucin degradation mechanism. These genes are also conserved in other *A. muciniphila* sp., including *A. muciniphila* CAG:154, *A. muciniphila* CAG:344 WG and *A. muciniphila* KLE1798, (Supplementary Fig. 10), suggesting that these enzymes could collaborate together in the specific degradation of mucins.”

9. A less important issue is that I think the first section of the results can be shortened a bit. There is a lot of information in there that is not central to the story and even hardcore structural biologists would find unnecessary, e.g. detailed descriptions of secondary structure, protein dimensions, space groups, etc. All that information is either visually evident in the figures, can be derived from the stats tables, or will ultimately be available when the PDBs are public.

ANSWER: Many thanks for the suggestion. However, we find useful that this information is described in the main text, reducing repeated visits to the Methods and Supplementary sections.

10. With all due respect to the authors, significant portions of this manuscript come across as an advertisement for a product. For example, the authors specifically refer to the OpeRATOR and GlycOCATCH products in the introduction, and 1-2 paragraphs in the discussion largely point out success stories of these products. I am not sure how appropriate this is, and I personally found it quite distracting from a very strong study that tells an otherwise interesting story. However, this is my opinion and not a catastrophic issue, so I will leave this to the discretion of the Editor.

ANSWER: Very honestly, it was not the goal to advertise the products. The analysis of O-glycan structures in peptide/proteins was/is still an important challenge in the field of glycosciences. For that reason, we have incorporated several successful examples on the use of OgpA enzyme. However, following the reviewer’s suggestion, we have made an effort to reduce (i) the reference to the commercial products in the Introduction section, and (ii) the amount of such information in the Discussion section.

Reviewer #2:

Trastoy et al have submitted a manuscript on the structure of a glycopeptide hydrolyzing protease from *Akkermansia muciniphila* including complexes with a glycopeptide substrate and a product derived thereof, respectively. This very interesting study also includes well comparison with structures of other related metalloproteases including such with known O-glycan specificity.

ANSWER: We very much appreciate that you consider our study interesting, and all the constructive comments/suggestions to improve the article.

The manuscript reads easily, although the figures can benefit from more labelling as detailed below. Moreover, it may be useful to make subheadings in the long Discussion.

ANSWER: Many thanks to point this to us. We have made a substantial effort to improve the grammar and readability of the entire manuscript, including subheadings in the Discussion section.

Some points to consider are listed below.

Major:

1. There seems to be claimed “essentially” no specificity on the peptide part for the O-glycan substrate. Can this be further explained? Can in fact “any sequence be accommodated”? How will specificity on the peptide part be seen in the light of the single substrate sequence analyzed in the present manuscript?

How many different peptide sequences will there be with relevant O-glycan units in the niche habitat of this bacterium?

ANSWER: OgpA showed proteolytic activity against two O-glycopeptides with different amino acid sequence: (i) a peptide based on MUC-1 (hydrolytic experiments) and glycodrosocin (X-ray crystallography). In that sense, and as we have mentioned in page 15:

“The crystal structure of OgpA_{H205A/D206A}-GD-SUB shows that protein-peptide interaction is primarily mediated by hydrogen bonds between the Pro-P3, Arg-P2, Pro-P1 and Ser-P1’ backbone and the residues of OgpA that form the corresponding subsites (Fig. 6). The side chains of these amino acid residues also interact with OgpA by hydrophobic interactions. Mucins, the potential target for OgpA *in vivo*, consist of a long, densely O-glycosylated domain with sequences rich in Pro, Thr, and Ser, often characterized by tandem repeats. His-P3’, Ser-P4, Tyr-P5, pro-P6 and Arg-P7 that form the O-glycopeptide are exposed to the solvent and do not interact with the protein. This indicates that the side chain nature of these residues is not a key element for OgpA activity.”

Therefore, we propose that only the neighboring amino acid to the O-glycosylated threonine might influence the activity of OgpA against mucins.

2. When the authors say (page 6) they will use the WT2 form for the description, then why is WT1 shown in Figure 1?

ANSWER: Many thanks to point this to us. It was a typo. We have replaced WT1 by WT2.

3. On page 6, describing the topology the beta-4 strand is mentioned to be antiparallel, so when the peptide substrate is mentioned elsewhere to be parallel, is this then to beta-4 or to the other four strands of the beta-sheet? (I assume the latter, but please clarify).

ANSWER: We agree with the reviewer. The beta-4 is parallel to the other four strands of the beta-sheet. We have clarified this issue accordingly:

“The GD substrate is inserted as an extra parallel β -strand, relative to the other four strands of the β -sheet comprised in the catalytic domain (Fig. 2f,g; Supplementary Fig. 4)”

One reason for clarifying can be that it would in fact be very helpful if this clear insertion of the substrate as a additional strand in the sheet can be illustrated on the relevant figures, maybe also the beta-strands or some of them can be numbered on relevant figures.

ANSWER: We agree with the reviewer. We have labelled additional β -strands on Figures 1, 2, and 3.

The insertion into the beta-sheet is not the impression one at the moment can obtain from the current figure 2, but I don't know if that will be simple to illustrate by molecular graphics?

ANSWER: We agree with the reviewer. We have incorporated a new Supplementary Figure 4 to illustrate the topology of the β -strands, and the insertion of the substrate peptide.

4. Can the authors help the readers by connecting M11, M12 and M60 families, perhaps even on page 7?

ANSWER: We agree with the reviewer. We have modified the text accordingly:

In page 7, we have included:

“Both families, M11 and M12, share the common motif HEXXHXXXXXH, in which the three His residues are zinc ligands and the Glu residue has a catalytic function. Structurally, the closest homologues also belong to the M12 family. However, it is worth noting that the M11 family is not structurally characterized. The M11 and M12 families belong to the MA(M) subclan, which comprises a “metzincin” fold, with a methionine C-terminal to the Zn^{2+} atom.^{63,65} Members of M11 and M12 families are proenzymes that require activation by limited proteolysis.”

In page 12, we have included:

“Members of the M60, M66 and M88 families belong to the MA clan. They share the common catalytic mechanism and the conserved metal-binding motif HEXXH described for OgpA (Supplementary Fig. 6). Members of the M60 and M88 families coordinate the Zn^{2+} atom with two histidine residues and one aspartic acid, while members of the M66 family do not show the conserved methionine residue.”

5. When it is mentioned that the M12 enzymes show no specificity against glycopeptides, does this mean that they are not able to cleave glycopeptides or that they can cleave these as well as non-glycosylated peptides, in other words these other enzymes just don't need the glycosylation in the substrates?

ANSWER: We agree with the reviewer. OgpA does not cleave non-glycosylated peptides. We have modified the text accordingly.

“and non-glycosylated peptides were not digested by OgpA.”

6. I assume that the full length GD was in the structure, but that density was not seen for the residues 1-3 and 14-19. One may read that it is a truncated form present in the structure, although admittedly the authors refer to “density”.

ANSWER: We agree with the reviewer. We have modified the text accordingly:

“We assume that the full length GD is in the crystal, but no density was observed for the residues 1-3 and 14-19, probably due to conformational flexibility.”

7. Which kind of electrostatic interaction do the authors have in mind between the 2-acetamido group on GalNAc and the O2 of Gal? (page 9).

ANSWER: Many thanks to point this to us. We have modified the text accordingly:

“Finally, the carbonyl oxygen of the 2-acetamido group of GalNAc forms a hydrogen bond with the O2 of the Gal residue.”

8. Which acetylation of the 6-position do the authors think of when comparing C1 and C3? (page 12).

ANSWER: Many thanks for point this to us. We have modified the text accordingly:

“In this case, the only differences compared to C1 are the arrangement of the 4-OH group that converts galactose to glucose and the acetylation of the 2-position of the GlcNAc residue (Fig. 4h).”

9. When the authors state that the specificity differs for the various O-glycan recognizing proteases with regard to the O-glycan recognition. Does this then consider that there can be prominent and poor substrates for these enzymes as demonstrated in the present work in the case of OgpA? One may ask if other another enzyme also prefers C1? And if that is the case then perhaps also have very low activity on the other substrates listed in the text. Moreover, in the HPLC chromatograms in the Supplemental Information where the asterisks are marking the position of the main product, will there be a trace in other candidate substrates tested than the two where the product of the poor activity is shown in a “blow-up”?

ANSWER: We have included in the text all O-glycopeptides that have been tested against the referred enzymes. It is worth noting that the reported data is qualitative. The authors do not extrapolate substrate preference information:

“*In vitro* experiments on defined synthetic O-glycopeptides (TnAg, C1, 6SC1 and 3SC1) revealed that IMPa is able to hydrolyze TnAg, C1, 6SC1 and 3SC1, whereas BT4244 hydrolyzes TnAg, and ZmpB exclusively hydrolyzes 6SC1.⁷¹”

Finally, according to the HPLC chromatograms reported in the Supplemental Information section, we could not identify any other substrates for OgpA than those described in the manuscript.

10. Although belonging to the Materials, it would be good to have proper names for the various glycoside hydrolases applied to generate the substrates rather than just the company name for the enzymes.

ANSWER: We agree with the reviewer. We have modified the text accordingly.

11. In Figure 1 it is a bit unclear what is the idea in panel b? That the “OCATCH” can select and hence purify an O-glycan peptide?

ANSWER: We agree with the reviewer. OCATCH can select and hence an O-glycan peptide.

There are shown structures from very many different view-angles, may it be possible to label for example alpha helix 3 on several figures and other central elements to help the readers? This includes figures in the Supplementary Information.

ANSWER: We have incorporated this information in the new version of the manuscript.

12. On Supplementary Figure 2 can some structural elements be tagged? Secondly are the peptide SUB and PRO not seen on the part of the molecule? (panels c and d). Is it beta-strand 4 which is yellow on panel c? And why is this?

ANSWER: By journal policy: “To facilitate assessment of the quality of the structural data, a stereo image of a portion of the electron density map (for crystallography papers) should be provided with the submitted manuscript.” The goal of Supplementary Figure 2 is to reflect the quality of the electron density maps.

13. How long reaction time was used for samples subjected to HPLC analysis in Supplemental Figure 6.

ANSWER: The reaction time was 18 hs. The information was incorporated in the legend of Supplemental Figure 6.

Minor:

1. In the abstract correct to “...A. muciniphila uses to...”.

ANSWER: We have modified the text accordingly.

2. In the abstract rephrase to “...that exclusively hydrolyzes the peptide bond...residues substituted with...” – If this is correctly understood by this Reviewer. (same point on page 5).

ANSWER: We have modified the text accordingly.

3. Page 3, correct to “..of fully processing a small...”.

ANSWER: We have modified the text accordingly.

4. Page 3, correct to “..depolymerization of glycan structures..”.

ANSWER: We have modified the text accordingly.

5. Page 3, correct two grammatical problems to “..contains polysaccharide utilization loci that encode highly..”.

ANSWER: We have modified the text accordingly.

6. Correct to Gram (not gram, this is named after a Danish bacteriologist).

ANSWER: We have modified the text accordingly.

7. Page 5, correct to “..specific digestion patterns that..”.

ANSWER: We have modified the text accordingly.

8. Page 8, correct to “..of an N-terminal catalytic..”.

ANSWER: We have modified the text accordingly.

9. Page 17, correct to “.., the latter acting..”.

ANSWER: We have modified the sentence.

10. D for carbohydrate stereo form should be in small cap.

ANSWER: We have modified the text accordingly.

11. Page 18, correct to “.. effort has been...”.

ANSWER: We have modified the text accordingly.

12. In the METHODS please refer to the SI where relevant for example on the production of the recombinant protein. Can the yields be informed on the recombinant proteins?

ANSWER: We have referred to Supplementary Figure 1 in the Methods section.

13. Page 23, last line correct to phosphate.

ANSWER: We have modified the text accordingly.

14. This Reviewer is in doubt what is meant by “..product and educt..” what is “educt”? (page 24)

ANSWER: We agree with the reviewer. In chemistry, an educt is a compound found in a reaction mixture as a starting material. It is not generated during the reaction process. We have replaced ‘educt’ by ‘substrate’, for clarity.

Reviewer #3: Prof. Rita Gerardy-Schahn.

The significance of a well-balanced microbiome for the individuals' health status is well documented, but the molecular mechanisms that establish the mutualistic interactions between host and microbiome are far from being understood. Nevertheless, the current knowledge allows to state that the complementary benefit arises to a large part via carbohydrate metabolism. The manuscript by Trastoy et al. focuses at this domain and presents the X-ray crystal structure of a microbial enzyme O-glycan protease (OgpA) from a key beneficial species (*Akkermansia muciniphila*) involved in the control of the gut barrier thickness and function. OgpA can degrade host mucins, a group of heavily O-glycosylated proteins that are essential in barrier integrity. OgpA cuts the protein backbone in dependence of O-glycans and this study reveals the molecular basis for the enzymatic mechanism and specificity. Crystals were obtained for the wild-type apo enzyme and in complex with the product. A functionally inactive variant (OgpA-H205A/E206A) was used to solve the substrate ligated structure. The combined data together with in vitro enzymatic studies on the 8 most common O-glycan variants and molecular docking studies allowed the detailed description of the enzymatic mechanism and revealed the basis for the enzymes' specificity. This is a well performed and comprehensive study, providing important novel data towards the understanding of the mutualistic host microbiome interactions and simultaneously fundamental mechanistic insight in an enzyme of utmost value in mucin analytics. In my eyes this study is ready for publication after a careful check with respect to grammatical issues.

ANSWER: We very much appreciate that you consider our study is well performed providing important novel data of the mutualistic host microbiome interactions and simultaneously fundamental mechanistic insight in an enzyme of utmost value in mucin analytics; and all the constructive comments/suggestions to improve the article.

There are numerous mistakes in verb forms (plural/singular); Gram (Hans Christian Gram 1853–1938) is a proper name and must be capitalized. There are more minor spelling errors that need to be addressed.

ANSWER: Many thanks to point this to us. We have made a substantial effort to improve the grammar and readability of the entire manuscript.

We wish to take this opportunity to thank the referees for her/his many thoughtful suggestions that have made the manuscript so much better.

REVIEWERS' COMMENTS:

Reviewer #1 (Remarks to the Author):

The authors have done a nice job of editing their manuscript to account for all of the reviewers' suggestions. My only remaining question is that the authors now seem somewhat contradictory regarding the classification of their protein. It is clearly revealed that OgpA has no significant overall amino acid sequence identity, other than the metal binding site, to other MEROPS families (I also performed this analysis and got identical results to the authors). On this basis, in my opinion, the NCBI annotation is very puzzling indeed as it seems impossible to reproduce it! The MEROPS families are established only on the basis of amino acid sequence identity (not tertiary structure - though this would come with significant amino acid sequence identity - and not function), which would then clearly place OgpA in a new family. Despite this, the authors state that more experimental and structural data would be required to establish a new family, which is not consistent with the sequence based classification of the families. I can understand the authors' potential reluctance to be equivocal on this. However, my feeling is that based on the evidence - no significant overall sequence identity to existing MEROPS families - the authors would not only be justified but are almost obligated to simply leave the argument at "Taken together, all the data suggests that OgpA could be considered as the founding member of a new family of peptidases."

Alisdair Boraston

Reviewer #2 (Remarks to the Author):

The authors have very well amended the manuscript adopting the points raised by the reviewers

Reviewer #3 (Remarks to the Author):

In my eyes this ms is ready for publication. An excellent study, indeed.

Reviewer #1: Prof. Alisdair Boraston

The authors have done a nice job of editing their manuscript to account for all of the reviewers' suggestions. My only remaining question is that the authors now seem somewhat contradictory regarding the classification of their protein. It is clearly revealed that OgpA has no significant overall amino acid sequence identity, other than the metal binding site, to other MEROPS families (I also performed this analysis and got identical results to the authors). On this basis, in my opinion, the NCBI annotation is very puzzling indeed as it seems impossible to reproduce it! The MEROPS families are established only on the basis of amino acid sequence identity (not tertiary structure - though this would come with significant amino acid sequence identity - and not function), which would then clearly place OgpA in a new family. Despite this, the authors state that more experimental and structural data would be required to establish a new family, which is not consistent with the sequence based classification of the families. I can understand the authors' potential reluctance to be equivocal on this. However, my feeling is that based on the evidence - no significant overall sequence identity to existing MEROPS families - the authors would not only be justified but are almost obligated to simply leave the argument at "Taken together, all the data suggests that OgpA could be considered as the founding member of a new family of peptidases."

ANSWER: Many thanks to point this to us. We have modified the text according to the reviewer's suggestion.